# Diff-Instruct: A Universal Approach for Transferring Knowledge From Pre-trained Diffusion Models

Weijian Luo[1],[*] Tianyang Hu[2],[†] Shifeng Zhang[2], Jiacheng Sun[2], Zhenguo Li[2], Zhihua Zhang[1]

[1]Peking University, [2]Huawei Noah's Ark Lab

## Abstract

Due to the ease of training, ability to scale, and high sample quality, diffusion models (DMs) have become the preferred option for generative modeling, with numerous pre-trained models available for a wide variety of datasets. Containing intricate information about data distributions, pre-trained DMs are valuable assets for downstream applications. In this work, we consider learning from pre-trained DMs and transferring their knowledge to other generative models in a data-free fashion. Specifically, we propose a general framework called Diff-Instruct to instruct the training of arbitrary generative models as long as the generated samples are differentiable with respect to the model parameters. Our proposed Diff-Instruct is built on a rigorous mathematical foundation where the instruction process directly corresponds to minimizing a novel divergence we call Integral Kullback-Leibler (IKL) divergence. IKL is tailored for DMs by calculating the integral of the KL divergence along a diffusion process, which we show to be more robust in comparing distributions with misaligned supports. We also reveal non-trivial connections of our method to existing works such as DreamFusion [54], and generative adversarial training. To demonstrate the effectiveness and universality of Diff-Instruct, we consider two scenarios: distilling pre-trained diffusion models and refining existing GAN models. The experiments on distilling pre-trained diffusion models show that Diff-Instruct results in state-of-the-art single-step diffusion-based models. The experiments on refining GAN models show that the Diff-Instruct can consistently improve the pre-trained generators of GAN models across various settings. Our official code is released through `https://github.com/pkulwj1994/diff_instruct`.

## 1 Introduction

Over the last decade, the field of deep generative models has made significant strides across various domains such as data generation [30, 32, 51, 53, 24, 54, 25, 34], density estimation [35, 7], image-editing [46, 8] and others. Notably, recent advancements in text-driven high-resolution image generation [60, 59, 58] have pushed the limits of using generative models for Artificial Intelligence Generated Content (AIGC). Behind the empirical success are fruitful developments of a wide variety of deep generative models, among which, diffusion models (DMs) are the most prominent. DMs leverage the diffusion processes and model the data across a wide spectrum of noise levels. Their ease of training, ability to scale, and high sample quality have made DMs the preferred option for generative modeling, with numerous pre-trained models available for a wide variety of datasets and applications. The trained DMs contain intricate information about the data distribution, making them valuable **assets** for downstream applications.

---

[*]Email: luoweijian@stu.pku.edu.cn.
[†]Corresponding to: Tianyang Hu (hutianyang1@huawei.com)

37th Conference on Neural Information Processing Systems (NeurIPS 2023).

Compared with training from scratch, extracting knowledge from a zoo of pre-trained models enables us to learn more efficiently. For instance, [15] employed a variety of pre-trained feature extractors and significantly boosted the state-of-the-art performance on domain generalization benchmarks. [9, 52, 16] exploited the rich multi-modal information stored in off-the-shelf CLIP models [55] for efficient text-guided image generation. Currently, we are witnessing a rising trend of *learning from models*, especially when accessing large amounts of high-quality data is difficult. Such a *model-driven* learning scheme can be particularly appealing for handling new tasks by providing a solid base model, which can be further improved by additional training data. While this research direction has been extensively investigated for discriminative models and supervised learning tasks [71, 38, 3, 72, 57, 56], its application to generative models remains largely unexplored. To this end, we are motivated to study the following question.

*(Q1): Can we transfer knowledge from pre-trained DMs to other generative models instead of learning from original training data?*

The seminal work DreamFusion [54] demonstrated the feasibility of such a quest for text-to-3D generation. Without a large text-labeled 3D dataset, [54] took advantage of the rich text-to-image knowledge stored in a large-scale diffusion model, the Imagen [23], to learn the 3D Neural Radiance Fields (NeRF) and achieved surprisingly good performance without using any 3D data. DreamFusion is a rare success and for general scenarios, the task can be difficult due to the vast difference among generative models. DMs represent a class of explicit generative models wherein the data's score function is modeled. Conversely, in various downstream applications, implicit generative models are favored due to their inherent flexibility and efficiency. An implicit model typically learns a neural transformation (i.e., a generator) that maps from a latent space to the data space, such as in generative adversarial networks (GANs), thereby enabling expeditious generation.

By exploring diverse architectural designs and latent space configurations, implicit models can readily adapt to structural constraints (e.g., molecules must be chemically valid, etc. [14, 10, 63]), assimilate prior knowledge [6, 13, 76, 45], and exhibit other advantageous properties. For implicit models that lack explicit score information, how to receive supervision from DM's multi-level score network is technically challenging, which greatly limits the potential use cases of pre-trained DMs. Therefore, we would like to further address the following question:

*(Q2): Can we tackle this challenge so that knowledge from DMs can be more broadly transferred?*

In this work, we give affirmative answers to (Q2) and propose a universal framework, **Diff-Instruct** (**DI**), to leverage pre-trained DMs to instruct the training of arbitrary implicit generative models as long as the generated samples from the implicit model are differentiable with respect to model parameters. When applied to single-step generation models such as GANs, Diff-Instruct provides an alternative non-adversarial training scheme. When the student model is a U-Net (with a fixed time), our method enters as a strong contender in the diffusion distillation literature [43, 61, 66], providing extreme acceleration for sampling from DMs with even one single step.

Our proposed Diff-Instruct is built on a rigorous mathematical foundation where the instruction process directly corresponds to minimizing a novel divergence we call Integral Kullback-Leibler (IKL) divergence. IKL is tailored for DMs by calculating the integral of the KL divergence along a diffusion process, which we show to be more robust in comparing distributions with misaligned supports (Section 3.2). We also reveal non-trivial connections of our method to existing works such as DreamFusion (Section 3.3.1) and generative adversarial training (Section 3.3.2). Interestingly, we show that the SDS objective can be seen as a special case of our Diff-Instruct on the scenario that the generator outputs a Dirac's Delta distribution.

To demonstrate the effectiveness and universality of Diff-Instruct, we consider two scenarios mentioned earlier: distilling pre-trained diffusion models to single step (Section 4.1) and improving pre-trained GAN generators (Section 4.2). The experiments on distilling pre-trained diffusion models on the ImageNet dataset of a resolution of $64 \times 64$ show that Diff-Instruct results in state-of-the-art single-step diffusion-based models over both diffusion distillation, such as the consistency distillation [66] and direct training methods [66, 83, 73]. The experiments on improving GAN generators show that the Diff-Instruct can consistently improve the pre-trained generators across various settings.

## 2   Preliminary

Assume we observe data from the underlying distribution $p_d(\boldsymbol{x})$. In generative modeling, we want to generate new samples $\boldsymbol{x} \sim p_d(\boldsymbol{x})$, where there are mainly two approaches, explicit and implicit. Currently, DMs are the most powerful explicit models while GANs are the most powerful implicit models.

**Diffusion models.**   The forward diffusion process of DM transforms any initial distribution $p^{(0)}$ towards some simple noise distribution,

$$d\boldsymbol{x}_t = \boldsymbol{F}(\boldsymbol{x}_t, t)\mathrm{d}t + G(t)d\boldsymbol{w}_t, \tag{2.1}$$

where $\boldsymbol{F}$ is a pre-defined drift function, $G(t)$ is a pre-defined scalar-value diffusion coefficient, and $\boldsymbol{w}_t$ denotes an independent Wiener process. A multiple-level or continuous-indexed score network $\boldsymbol{s}_\phi(\boldsymbol{x}, t)$ is usually employed in order to approximate marginal score functions of the forward diffusion process (2.1). The learning of marginal score functions is achieved by minimizing a weighted denoising score matching objective [69, 65],

$$\mathcal{L}_{DSM}(\phi) = \int_{t=0}^{T} w(t)\mathbb{E}_{\boldsymbol{x}_0 \sim p^{(0)}, \boldsymbol{x}_t | \boldsymbol{x}_0 \sim p_t(\boldsymbol{x}_t | \boldsymbol{x}_0)} \| \boldsymbol{s}_\phi(\boldsymbol{x}_t, t) - \nabla_{\boldsymbol{x}_t} \log p_t(\boldsymbol{x}_t | \boldsymbol{x}_0) \|_2^2 \mathrm{d}t. \tag{2.2}$$

Here the weighting function $w(t)$ controls the importance of the learning at different time levels and $p_t(\boldsymbol{x}_t | \boldsymbol{x}_0)$ denotes the conditional transition of the forward diffusion (2.1). High-quality samples from a DM can be drawn by simulating SDE which is implemented by learned score network [65]. However, the simulation of an SDE is significantly slower than that of other models such as implicit models.

**Generative adversarial networks.**   GANs are representative implicit generative models [30, 75, 67, 77]. They leverage neural networks (generators) to map an easy-to-sample latent vector to generate a sample. Therefore they are efficient. However, the training of GANs is challenging, particularly because of the reliance on adversarial training. To train a GAN model, a neural discriminator $h$ is optimized to distinguish the data and generated samples. This leads to the creation of a surrogate probability metric $D_h(\cdot, \cdot)$ between $p_g(\boldsymbol{x})$ and $p_d(\boldsymbol{x})$. The generator is updated based on this metric, with the aim of improving the quality of the generated samples [19, 2, 44]. The objective of a most commonly used GAN [19] can be written as

$$\mathcal{L}_h = -\mathbb{E}_{x \sim p_d}[\log h(\boldsymbol{x})] - \mathbb{E}_{z \sim p_z}[\log(1 - h(g(\boldsymbol{z})))], \quad \mathcal{L}_g = -\mathbb{E}_{z \sim p_z}[\log h(g(\boldsymbol{z}))],$$

where the training alternates between minimizing $\mathcal{L}_h$ and $\mathcal{L}_g$ with the other part fixed. For a fixed $g$, the optimal $h$ should recover the density ratio $p_d(\boldsymbol{x})/(p_d(\boldsymbol{x}) + p_g(\boldsymbol{x}))$, and in turn, $D_h$ is the Jensen-Shannon divergence. There are variants of objectives that minimize other divergences. For instance, if the $\mathcal{L}_g^{(KL)} = \mathbb{E}_{z \sim p_z}[\log \frac{1 - h(g(\boldsymbol{z}))}{h(g(\boldsymbol{z}))}]$, the objective aims to minimize the KL divergence between generator and data distribution. In Section 3.3.2, we establish the equivalence of our Diff-Instruct with the adversarial training that aims to minimize the KL divergence.

**Neural radiance fields.**   The neural radius field (NeRF) [48] is a kind of 3D object model that uses a multi-layer-perceptron (MLP) to map coordinates of a mesh grid to volume properties such as color and density. Given the camera parameters, a rendering algorithm can output a 2D image that is a view projection of the 3D NeRF. The rendering algorithm is usually differentiable to learnable parameters of NeRF's MLP, this makes the NeRF can be updated through proper instructions on the rendered 2D image.

## 3   Diff-Instruct

The main goal of Diff-Instruct is to transfer the knowledge of a pre-trained DM to other generative models. To demonstrate the universality of our approach, we consider the more general case where the student model is an implicit model, i.e., a generator.

**Problem setup.**   Recall our setting that we have a pre-trained diffusion model with the multi-level score net denoted as $\boldsymbol{s}_{p^{(t)}}(\boldsymbol{x}_t) := \nabla_{\boldsymbol{x}_t} \log p^{(t)}(\boldsymbol{x}_t)$ where $p^{(t)}(\boldsymbol{x}_t)$'s are the underlying distributions diffused at time $t$ according to (2.1). Assume the pre-trained diffusion model provides a sufficiently good approximation of data distribution, i.e., $p^{(0)} \approx p_d$. For ease of mathematical treatment, we use

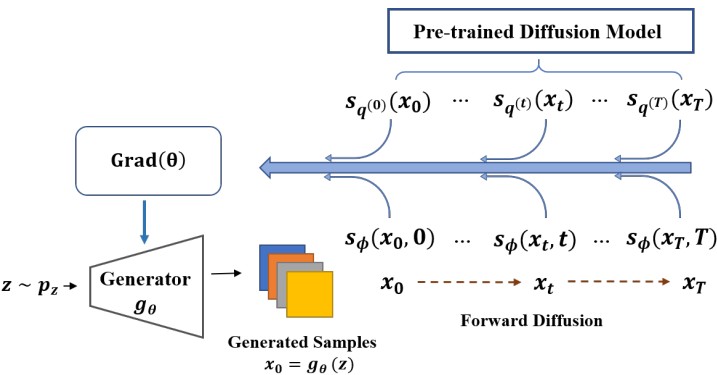

Figure 1: Illustration of our *Diff-Instruct* pipeline. The generator accepts instructions from all diffusion time levels to calculate the gradient of the Integral KL divergence. The gradient is used to update its parameters. This data-free learning scheme enables us to employ pre-trained DMs as teachers to instruct a wide variety of generative models.

$p^{(0)}, p_d$ interchangeably. The goal of our Diff-Instruct is to train an implicit model $g_\theta$ without any training data, such that the distribution of the generated samples, denoted as $p_g$, matches that of the pre-trained DM. The instruction process involves minimizing certain probability divergences between the implicit distribution and the data distribution.

**Instruction criterion.** In order to receive supervision from the multi-level score functions $\boldsymbol{s}_{p^{(t)}}(\boldsymbol{x}_t)$, introducing the same diffusion process to the generated samples seems inevitable. Consider diffusing $p_g$ along the same forward process as the instructor DM and let $q^{(t)}$ be the corresponding densities at time $t$. Let $\boldsymbol{s}_{q^{(t)}}(\boldsymbol{x}_t) := \nabla_{\boldsymbol{x}_t} \log q^{(t)}(\boldsymbol{x}_t)$ be the marginal score functions. At each time level, how to design the instruction criterion and how to combine different time levels are of critical importance. To this end, we consider integrating the Kullback-Leibler divergence along the forward diffusion process with a proper weighting function, such as in (2.2). The resulting Integral Kullback-Leibler (IKL) divergence is a valid probability divergence with two important properties: 1) IKL is more robust than KL in comparing distributions with misaligned supports; 2) The gradient of IKL with respect to the generator's parameters only requires the marginal score functions of the diffusion process, making it a suitable divergence for incorporating the scoring network of pre-trained diffusion models.

In the following sections, we first formally define the IKL and then go into detail about the mathematical ground of our Diff-Instruct algorithms. We then establish the connections of Diff-Instruct to existing methods and discuss in detail a novel application of Diff-Instruct on data-free diffusion distillation, together with a comparison to existing diffusion distillation approaches.

## 3.1 Integral Kullback-Leibler divergence

The IKL is tailored to incorporate knowledge of pre-trained DMs in multiple time levels. It generalizes the concept of KL divergence to involve all time levels of the diffusion process.

**Definition 3.1** (Integral KL divergence). Given a diffusion process (2.1) and a proper weighting function $w(t) > 0, t \in [0, T]$, the IKL divergence between two distributions $p, q$ is defined as

$$\mathcal{D}_{IKL}^{[0,T]}(q, p) := \int_{t=0}^{T} w(t) \mathcal{D}_{KL}(q^{(t)}, p^{(t)}) \mathrm{d}t := \int_{t=0}^{T} w(t) \mathbb{E}_{\boldsymbol{x}_t \sim q^{(t)}} \Big[ \log \frac{q^{(t)}(\boldsymbol{x}_t)}{p^{(t)}(\boldsymbol{x}_t)} \Big] \mathrm{d}t, \qquad (3.1)$$

where $q^{(t)}$ and $p^{(t)}$ denote the marginal densities of the diffusion process (2.1) at time $t$ initialized with $q^{(0)} = q$ and $p^{(0)} = p$ respectively.

The IKL divergence integrates the KL divergence along a diffusion process, which enables us to amalgamate knowledge from pre-trained DM at multiple diffusion times. For simplicity, we use the notation $\mathcal{D}_{IKL}(p, q)$ to represent $\mathcal{D}_{IKL}^{[0,\infty)}(p, q)$ if the integral exists. Since the KL divergence is well-defined, the proposed IKL divergence, as the integral of KL divergence, is also well-defined.

**Proposition 3.2.** The $\mathcal{D}_{IKL}(q, p)$ satisfies that $\mathcal{D}_{IKL}(q, p) \geq 0, \forall q, p$. Furthermore, the equality holds if and only if $q = p$, almost everywhere under measure $p$.

---

**Algorithm 1:** Diff-Instruct Algorithm

---

**Input:** pre-trained DM $\boldsymbol{s}_{p^{(t)}}$, generator $g_\theta$, prior distribution $p_z$, DM $\boldsymbol{s}_\phi$; forward diffusion (2.1).

**while** *not converge* **do**

    update $\phi$ using SGD with gradient

$$\text{Grad}(\phi) = \frac{\partial}{\partial\phi} \int_{t=0}^T w(t) \mathbb{E}_{\substack{\boldsymbol{z}\sim p_z, \boldsymbol{x}_0 = g_\theta(\boldsymbol{z}), \\ \boldsymbol{x}_t | \boldsymbol{x}_0 \sim p_t(\boldsymbol{x}_t | \boldsymbol{x}_0)}} \|\boldsymbol{s}_\phi(\boldsymbol{x}_t, t) - \nabla_{\boldsymbol{x}_t} \log p_t(\boldsymbol{x}_t | \boldsymbol{x}_0)\|_2^2 \, \mathrm{d}t.$$

    update $\theta$ using SGD with the gradient

$$\text{Grad}(\theta) = \int_{t=0}^T w(t) \mathbb{E}_{\substack{\boldsymbol{z}\sim p_z, \boldsymbol{x}_0 = g_\theta(\boldsymbol{z}), \\ \boldsymbol{x}_t | \boldsymbol{x}_0 \sim p_t(\boldsymbol{x}_t | \boldsymbol{x}_0)}} \left[\boldsymbol{s}_\phi(\boldsymbol{x}_t, t) - \boldsymbol{s}_{p^{(t)}}(\boldsymbol{x}_t)\right] \frac{\partial \boldsymbol{x}_t}{\partial \theta} \, \mathrm{d}t.$$

**end**
**return** $\theta, \phi$.

---

One of the advantages of using IKL instead of KL is its robustness. For instance, with proper weighting function, the IKL is well-defined even when the vanilla KL divergence degenerates to infinity. This demonstrates that the IKL divergence is more robust than the KL divergence for two distributions with misaligned supports. We consider a famous example in [2] where the generator distribution $p_g$ and target distribution $p_d$ have disjoint support. In this case, the KL divergence between $p_g$ and $p_d$ degenerates to positive infinity, while the IKL divergence has a finite value for all generator parameters and unique minima that match the generator and data distribution. Check Appendix A.1 for details.

### 3.2 Instruct algorithm

Let $g_\theta$ be the generator of the implicit model. Let $q^{(0)}$ denote the implicit distribution for samples which are obtained by $\boldsymbol{x}_0 = g_\theta(\boldsymbol{z}), \boldsymbol{z} \sim p_z$ and denote $q^{(t)}$ as the marginal distribution of the forward SDE ((2.1)) initialized with $q^{(0)}$. Let $\{p^{(t)}\}_{t\in[0,\infty]}$ and $\{\boldsymbol{s}_{p^{(t)}}(.)\}_{t\in[0,\infty]}$ represent the marginal densities and score functions of the pre-trained diffusion model. Our Diff-Instruct aims to minimize the IKL between $q^{(0)}$ and $p^{(0)}$ so as to update the generator's parameters. Following the notations of definition (3.1), we give a non-trivial gradient formula for minimizing the IKL in Theorem 3.3 that includes only the score functions.

**Theorem 3.3.** The gradient of the IKL in (3.1) between $q^{(0)}$ and $p^{(0)}$ is

$$\text{Grad}(\theta) = \int_{t=0}^T w(t) \mathbb{E}_{\substack{\boldsymbol{z}\sim p_z, \boldsymbol{x}_0 = g_\theta(\boldsymbol{z}), \\ \boldsymbol{x}_t | \boldsymbol{x}_0 \sim p_t(\boldsymbol{x}_t | \boldsymbol{x}_0)}} \left[\boldsymbol{s}_\phi(\boldsymbol{x}_t, t) - \boldsymbol{s}_{p^{(t)}}(\boldsymbol{x}_t)\right] \frac{\partial \boldsymbol{x}_t}{\partial \theta} \, \mathrm{d}t. \tag{3.2}$$

Theorem 3.3 gives an explicit gradient to minimize the IKL divergence w.r.t. the parameter of the generator. Note that the gradient estimation only requires the marginal score functions $\boldsymbol{s}_{p^{(t)}}$ and $\boldsymbol{s}_{q^{(t)}}$. If the marginal score functions of the implicit distribution can be approximated by another diffusion model $\boldsymbol{s}_\phi(\boldsymbol{x}_t, t)$, we can utilize the gradient formula (3.2) to update the generator's parameter $\theta$.

Now we formally propose *Diff-Instruct* as in Algorithm 1, which trains the implicit model through two alternative phases between learning the marginal score functions $\boldsymbol{s}_\phi$, and updating the implicit model with gradient (3.2). The former phase follows the standard DM learning procedure, i.e., minimizing loss function (2.2), with a slight change that the data is generated from the generator. The resulting $\boldsymbol{s}_\phi(\boldsymbol{x}_t, t)$ provides an estimation of $\boldsymbol{s}_{q^{(t)}}(\boldsymbol{x}_t)$. The latter phase updates the generator's parameter $\theta$ using gradient from (3.2), where two needed functions are provided by pre-trained DM $\boldsymbol{s}_{p^{(t)}}(\boldsymbol{x}_t)$ and learned DM $\boldsymbol{s}_\phi(\boldsymbol{x}_t, t)$. When the algorithm converges, $\boldsymbol{s}_\phi(\boldsymbol{x}_t, t) \approx \boldsymbol{s}_{p^{(t)}}(\boldsymbol{x}_t)$, and the gradient (3.2) is approximately zero.

### 3.3 Connections to existing methods

In this section, we establish the connections of Diff-Instruct to two typical methods, the score distillation sampling proposed in DreamFusion [54], and the generative adversarial training in Goodfellow et al. [19].

### 3.3.1 Connection to score distillation sampling

The score distillation sampling (SDS) algorithm was proposed by Poole et al. [54] to distill the knowledge of a large-scale text-to-image diffusion model into a 3D NeRF model. The idea of SDS has been applied in various contexts, including the text-to-3D NeRF generation based on text-to-2D diffusion models [54, 39, 47], and the text-guided image editing [20].

It turns out that the SDS algorithm is a special case of our Diff-Instruct when the generator's output is a Dirac's Delta distribution with learnable parameters. More precisely, we find that Diff-Instruct's gradient formula (3.2) will degenerate to the gradient formula of SDS under the assumption that the generator outputs a Delta distribution.

**Corollary 3.4.** If the generator's output is a Dirac's Delta distribution with learnable parameters, i.e. $q(\boldsymbol{x}_0) = \delta_{g(\theta)}(\boldsymbol{x}_0)$ [3]. Then the gradient formula (3.2) becomes

$$\mathrm{Grad}(\theta) = \int_{t=0}^{T} w(t) \mathbb{E}_{\substack{\boldsymbol{x}_0 = g(\theta), \\ \boldsymbol{x}_t | \boldsymbol{x}_0 \sim p_t(\boldsymbol{x}_t | \boldsymbol{x}_0)}} \left[ \nabla_{\boldsymbol{x}_t} \log p_t(\boldsymbol{x}_t | \boldsymbol{x}_0) - \boldsymbol{s}_{p^{(t)}}(\boldsymbol{x}_t) \right] \frac{\partial \boldsymbol{x}_t}{\partial \theta} \mathrm{d}t. \tag{3.3}$$

(3.3) does not depends on another diffusion model $\boldsymbol{s}_{q^{(t)}}$ as in (3.2). So under the assumption of Corollary 3.4, there is no need for using another DM to estimate the generator's marginal score functions. This is because when the generator outputs a Delta distribution, there is no randomness in $\boldsymbol{x}_0$. So the marginal score functions are only determined by $p_t(\boldsymbol{x}_t | \boldsymbol{x}_0)$. The gradient (3.3) is equivalent to the score distillation sampling proposed in DreamFusion [54].

The fact that SDS is a special case of Diff-Instruct is not a coincidence, as in DreamFusion, the rendered image of a NeRF model from a certain camera view is a 2D image that is differentiable to NeRF's parameters. Therefore, using SDS to learn a NeRF model is essentially an application of using Diff-Instruct to distill a pre-trained text-to-2D diffusion model in order to obtain a 3D NeRF object. However, the path we obtain (3.3) is totally different from that in DreamFusion. In DreamFusion, the authors obtained (3.3) by taking the data gradient of the diffusion model's loss function (2.2) and *empirically* omitted the Jacobian term of the pre-trained score network. However, in this work, we first propose the general formulation of Diff-Instruct and then specialize it to obtain SDS in a natural way.

### 3.3.2 Connection to GANs

Our proposed Diff-Instruct without integral on time is equivalent to the adversarial training [19] that aims to minimize the KL divergence. Following the same notation as in Section 2, the adversarial training uses a discriminator $h(.)$ to learn the density ratio to construct the objective for the generator.

**Corollary 3.5.** If the discriminator $h$ learns the perfect density ratio, i.e. $h(\boldsymbol{x}) = \frac{p_d(\boldsymbol{x})}{p_d(\boldsymbol{x}) + p_g(\boldsymbol{x})}$, then updating the generator to minimize the KL divergence ($\mathcal{L}_g^{(KL)}$ in Section 2) is equivalent to Diff-Instruct with a weighting function $w(0) = 1$ and $w(t) = 0, \forall t > 0$.

The Diff-Instruct is essentially a different method from adversarial training in three aspects. First, the adversarial training relies on a discriminator network to learn the density ratio between the model distribution and data distribution. However, Diff-Instruct employs DMs instead of discriminators to instruct the generator updates. Second, in scenarios where only a pre-trained diffusion model is available without any real data samples, Diff-Instruct can distill knowledge from the pre-trained model to the implicit generative model, which is not achievable with adversarial training. Third, Diff-Instruct uses the IKL as the minimization divergence, which overcomes the degeneration problem of the KL divergence via a novel use of diffusion processes and can potentially overcome the drawbacks such as mode-drop issues of adversarial training.

## 3.4 Related works

The pre-trained diffusion models on large-scale datasets contain rich knowledge of the data distribution. There is a growing interest in distilling this knowledge to other models [43] that are more sampling-efficient, such as implicit generators [50, 26] and neural radiance fields models [48, 54].

One significant advantage of our Diff-Instruct framework is its ability to update the generator without real data $\boldsymbol{x} \sim p_d$. Instead, the knowledge of the data distribution to update the generator is contained

---

[3] We switch the notation from $g_\theta(z)$ to $g(\theta)$ since under the assumptions the generator has no randomness.

in the marginal score function $\boldsymbol{s}_{p^{(t)}}$ as in (3.2). This enables our Diff-Instruct to distill knowledge from pre-trained DMs into flexible generators in a data-free manner and provides a major advantage over other diffusion distillation methods that require either the real data or synthetic data from pre-trained DMs. More precisely, There are three kinds of diffusion distillation methods depending on how to use the data for distillation. The first is the *data synthetic distillation*, which requires using the pre-trained DM to synthesize data from random noises. The student model then learns the mechanism between random noise and synthetic data in order to enhance the efficiency of data generation. Representative methods of data-synthetic distillation are Knowledge Distillation (KD [42]), Rectified Flow (ReFlow [40]) and DFNO ([82]). The second method does not require synthetic data from diffusion models but involves real data when distilling. The consistency distillation (CD [66]) is a representative method. The third method is pure *data-free distillation*, which requires neither real data nor synthetic data. The Diff-Instruct and the Propgressive Distillation (PD [61]) are representative pure data-free distillation. Generally, we use the word "data-free distillation" to include both data-synthetic distillation and pure data-free distillation. To give a comprehensive comparison of diffusion distillation methods, we summarize three main features of diffusion distillation methods in Table 6. The efficiency represents the computational cost of the distillation method. Methods that require synthesizing dataset is inefficient. The data-synthetic distillation requires simulations with pre-trained DMs, thus is inefficient. The flexibility represents whether the distillation approach is capable of distilling knowledge of pre-trained DM to flexible generator architectures, for example, the generator whose input and output dimension is different. Diff-Instruct is the only method that can apply to a wide variety of downstream generators.

Furthermore, Diff-Instruct offers very high flexibility to the generator, distinguishing it from traditional diffusion distillation methods that impose strict constraints on the generator selection. For instance, the generator can be a convolutional neural network (CNN)-based or a Transformer-based image generator such as StyleGAN [28, 30, 31, 37], or an UNet-based generator [73] adapted from pre-trained diffusion models [32, 65, 22]. The versatility of Diff-Instruct allows it to be adapted to different types of generators, expanding its applicability across a wide range of generative modeling tasks. In the experiment sections, we show that Diff-Instruct is capable of transferring knowledge to generator architectures including both UNet-based and GAN generators respectively. To the best of our knowledge, the Diff-Instruct is the first approach to efficiently enable such a data-free knowledge transfer from diffusion models to generic implicit generators.

## 4    Experiments

With the abundance of powerful pre-trained DMs with diverse expertise, our proposed Diff-Instruct unlocks their potential as sources of knowledge to instruct a wide variety of models. To demonstrate, we choose the state-of-the-art DMs from the seminal work by [32] (we denote as EDMs) as instructors and consider transferring their knowledge to implicit generators. In this section, we evaluate the efficacy of DI through two downstream applications: diffusion distillation and improvement of GAN's generator. These two experiments correspond to using UNet and GAN's generator to absorb the knowledge from DMs. In the diffusion distillation experiments, we use Diff-Instruct to distill pre-trained DMs to single-step generative models. On the ImageNet $64 \times 64$ dataset, our Diff-Instruct achieves state-of-the-art performance in terms of FID among all single-step diffusion-based generative models. Furthermore, in the GAN-improving experiments, we use Diff-Instruct to improve existing GAN models that are pre-trained to convergence with adversarial training. The Diff-Instruct is shown to be able to consistently enhance the generative performance of pre-trained StyleGAN-2 models on the CIFAR10 dataset by incorporating knowledge of pre-trained DMs.

### 4.1    Single-step diffusion distillation

Diffusion distillation is a hot research area that aims to accelerate the generation speed of diffusion models. In our experiments, we utilize our Diff-Instruct framework to train single-step generators on CIFAR-10 [36] and ImageNet $64 \times 64$ [11] from pre-trained EDM [32] models. We evaluate the performance of the trained generator via Frechet Inception Distance (FID) [21], the lower the better, and Inception Score (IS) [62]), the higher the better. For additional details about the generator's architecture, pre-trained models, and the hyper-parameters on our experimental setup, please refer to Appendix B.1.

**Performances.**    Table 1 and 2 summarize the FID and IS of the single-step generator that we trained with Diff-Instruct from pre-trained EDMs on the CIFAR10 datasets (unconditional without labels)

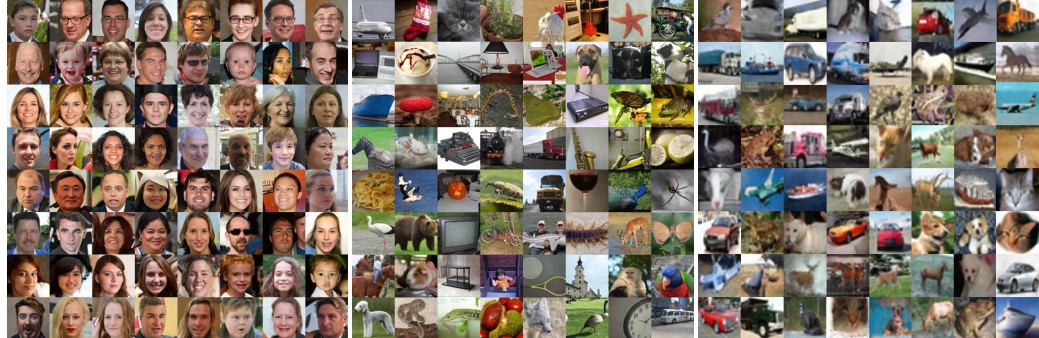

Figure 2: Generated samples from one-step generators that are distilled from pre-trained diffusion models on different datasets. *Left*: FFHQ-64 (unconditional); *Mid*: ImageNet-64 (conditional); *Right*: CIFAR-10 (unconditional).

Table 1: Unconditional sample quality on CIFAR-10 through diffusion generations. *Methods that require synthetic data construction for distillation. †Methods that require real data for distillation.

| METHOD | NFE (↓) | FID (↓) | IS (↑) |
|---|---|---|---|
| **Multiple Steps (include Diffusion Distillation)** | | | |
| DDPM [22] | 1000 | 3.17 | 9.46 |
| LSGM [68] | 147 | 2.10 | |
| PFGM [74] | 110 | 2.35 | 9.68 |
| EDM [32] | 35 | **1.97** | |
| DDIM [64] | 50 | 4.67 | |
| DDIM [64] | 10 | 8.23 | |
| DPM-solver-2 [41] | 12 | 5.28 | |
| DPM-solver-3 [41] | 12 | 6.03 | |
| 3-DEIS [78] | 10 | 4.17 | |
| UniPC [80] | 8 | 5.10 | |
| UniPC [80] | 5 | 23.22 | |
| Denoise Diffusion GAN(T=2) [73] | 2 | 4.08 | **9.80** |
| PD [61] | 2 | 5.58 | 9.05 |
| CT [66] | 2 | 5.83 | 8.85 |
| CD† [66] | 2 | 2.93 | 9.75 |
| **Single Step** | | | |
| Denoise Diffusion GAN(T=1) [73] | 1 | 14.6 | 8.93 |
| KD* [42] | 1 | 9.36 | |
| TDPM [83] | 1 | 8.91 | 8.65 |
| 1-ReFlow [40] | 1 | 378 | 1.13 |
| CT [66] | 1 | 8.70 | 8.49 |
| 1-ReFlow (+distill)* [40] | 1 | 6.18 | 9.08 |
| 2-ReFlow (+distill)* [40] | 1 | 4.85 | 9.01 |
| 3-ReFlow (+distill)* [40] | 1 | 5.21 | 8.79 |
| PD [61] | 1 | 8.34 | 8.69 |
| CD-L2† [66] | 1 | 7.90 | |
| CD-LPIPS† [66] | 1 | **3.55** | 9.48 |
| **Diff-Instruct** | 1 | 4.53 | **9.89** |

Table 2: Class-conditional sample quality on CIFAR-10 and ImageNet $64 \times 64$ through diffusion generations. *Methods that require synthetic data construction for distillation. †Methods that require real data for distillation.

| METHOD | NFE (↓) | FID (↓) |
|---|---|---|
| **Multiple Steps (include Diffusion Distillation)** | | |
| EDM [32] | 35 | **1.79** |
| EDM-Heun [32] | 20 | 2.54 |
| EDM-Euler [32] | 20 | 6.23 |
| EDM-Heun [32] | 10 | 15.56 |
| **Single Step** | | |
| EDM [32] | 1 | 314.81 |
| **Diff-Instruct** | 1 | **4.19** |

| **Class-conditional ImageNet $64 \times 64$.†Distillation techniques.** | | |
|---|---|---|
| METHOD | NFE (↓) | FID (↓) |
| **Multiple Steps** | | |
| ADM [12] | 250 | **2.07** |
| SN-DDIM [4] | 100 | 17.53 |
| EDM [32] | 79 | 2.44 |
| EDM-Heun[32] | 10 | 17.25 |
| GGDM [70] | 25 | 18.4 |
| CT [66] | 2 | 11.1 |
| PD† [61] | 2 | 8.95 |
| CD† [66] | 2 | 4.70 |
| **Single Steps** | | |
| EDM[32] | 1 | 154.78 |
| PD† [61] | 1 | 15.39 |
| CT [66] | 1 | 13.00 |
| CD-L2† [66] | 1 | 12.10 |
| CD-LPIPS† [66] | 1 | 6.20 |
| **Diff-Instruct** | 1 | **5.57** |

and the conditional generation on the ImageNet $64 \times 64$ data. Diff-Instruct performs competitively across all datasets among single-step and multiple-step diffusion-based generative models, which involve both models from diffusion distillation or direct training.

As shown in Table 2, on the ImageNet dataset of the resolution of $64 \times 64$, Diff-Instruct outperforms diffusion-based single-step generative models in terms of FID, including both distillation methods that require real data or synthetic data, and even models that are trained from scratch. On the unconditional generation of the CIFAR10 dataset, Diff-Instruct achieves the state-of-the-art IS among diffusion-based single-step generative models but achieves the second-best FID, only worse than the consistency distillation (CD) [66] which requires both real data for distillation and the learned neural image metric (e.g. LPIPS [79]). The conditional generation experiment on the CIFAR10 dataset shows that the Diff-Instruct performs better than a 20-NFE diffusion sampling from EDM model [32] with Euler–Maruyama discretization but worse than a 20-NEF Heun discretization.

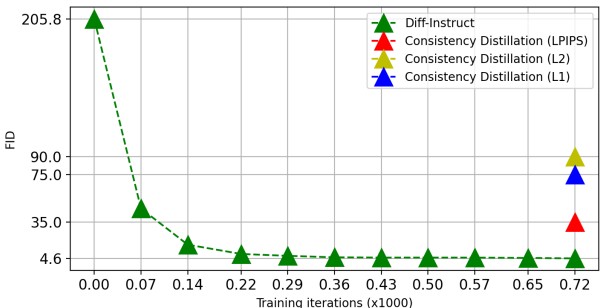

Figure 3: Comparison of FID convergence on diffusion distillation of CIFAR10 unconditional EDM model [32]. The FID value of consistency distillation is obtained from its original paper [66].

Figure 2 shows some non-cherry-picked generated samples from a single-step generator trained with Diff-Instruct on the FFHQ [28], ImageNet [11], and the CIFAR10 [36] datasets of the resolution of $64 \times 64$. In conclusion, our Diff-Instruct can achieve competitive distillation performance under the most challenging conditions with no synthetic or real datasets. We put more discussions and analyses in the Appendix B.1.

**Remark 4.1.** In Table 1 and Table 2, PD, CD, and Diff-Instruct all use the EDM teacher, and the same UNet student with the same architecture as the teacher model (The author of consistency distillation re-implemented the PD for EDM, so the reported FID for PD is lower than that in PD's original paper). The Diff-Instruct generator uses the same UNet architecture as the teacher diffusion model, so the number of sampling steps (NFE) represents its inference time costs. As we show in the upper part of Table 2, sampling from our learned one-step generator with 1 NFE results in an FID of 4.19, which is significantly better than its teachers with 10 NFEs with an FID of 15.56. So we conclude that the one-step generator trained with Diff-Instruct achieves at least 10+ times acceleration (more efficient) than its teacher diffusion model.

**Fast convergence speed.** Another advantage of applying Diff-Instruct for diffusion distillation is the fast convergence speed. We empirically find that Diff-Instruct has a much faster convergence speed than other distillation methods and has a tolerance for a large learning rate for optimization. In Figure 3, we show the convergence of FID with respect to the optimization iterations of the generator trained on an unconditional DM on the CIFAR10 dataset. We set the optimization step size of Diff-Instruct to be $1e-4$ and the FID of the generator trained with Diff-Instruct converges fast within 7k iterations. However, the FID of the distilled diffusion model with consistency distillation algorithms does not converge with less than 7k iterations. One possible reason for the fast convergence speed is that Diff-Instruct's student model is a one-step generator that does not need to take multiple-time indexes in contrast to student models of other distillation methods such as CD and PD. This makes Diff-Instruct efficient when distillation without the need for learning at multiple time levels.

## 4.2   Improving generative adversarial networks

Another application of Diff-Instruct is to improve the generator of GAN models that are pre-trained with adversarial training 3.3.2.

**Experiement settings.** We take the pre-trained EDM model [32] on the CIFAR10 datasets as the instructor and the pre-trained StyleGAN-2 [29] models that are assumed to converge with adversarial training under different settings (conditional or unconditional with different data augmentation strategies). Our goal is to use pre-trained DMs to further improve pre-trained generators. We initialize the generator in the Diff-Instruct algorithm with the pre-trained StyleGAN-2 generator and initialize DMs for implicit distributions with the pre-trained EDM models. We then use Diff-Instruct to update the generator. We put more details in Appendix B.2.

**Performance.** As shown in Table 3 and 4, our Diff-Instruct can consistently improve the pre-trained generator's performance in terms of FID. More precisely, the FID of the pre-trained StyleGAN-2 with Adaptive Data Augmentation (ADA) is improved from 2.42 to 2.27 for the conditional setting and

Table 3: Class-conditional sample quality on CIFAR10 through GAN models. [†] Models that we implemented.

| METHOD | FID ($\downarrow$) | IS ($\uparrow$) |
|---|---|---|
| BigGAN [5] | 14.73 | 9.22 |
| BigGAN+Tune [5] | 8.47 | $9.07 \pm 0.13$ |
| MultiHinge [33] | 6.40 | $9.58 \pm 0.09$ |
| FQ-GAN [81] | 5.59 | $8.48 \pm 0.09$ |
| Stylegan2 [30] | 6.96 | $9.53 \pm 0.06$ |
| Stylegan2[†] | 7.01 | $9.23 \pm 0.07$ |
| **Stylegan2[†] + DI** | 6.62 | $9.40 \pm 0.06$ |
| Stylegan2+ADA [29] | 3.49 | **10.24** $\pm 0.07$ |
| Stylegan2+ADA+Tune [29] | 2.42 | $10.14 \pm 0.09$ |
| **Stylegan2+ADA+Tune + DI** | **2.27** | $10.11 \pm 0.10$ |

Table 4: Unconditional sample quality on CIFAR10 through GAN models. [†] Models that we implemented.

| METHOD | FID ($\downarrow$) | IS ($\uparrow$) |
|---|---|---|
| SNGAN [49] | 21.70 | 8.22 |
| ProGAN [17] | 15.52 | $8.56 \pm 0.06$ |
| AutoGAN [18] | 12.42 | $8.55 \pm 0.10$ |
| SNGAN+DGflow [1] | 9.35 | 9.62 |
| TransGAN [27] | 9.02 | 9.26 |
| StyleGAN2 [30] | 8.32 | $9.21 \pm 0.09$ |
| StyleGAN2[†] | 8.21 | $9.09 \pm 0.09$ |
| **StyleGAN2[†] + DI** | 7.56 | $9.16 \pm 0.09$ |
| StyleGAN2+ADA [29] | 5.33 | $10.02 \pm 0.07$ |
| StyleGAN2+ADA+Tune [29] | 2.92 | $9.83 \pm 0.04$ |
| **StyleGAN2+ADA+Tune + DI** | **2.71** | **9.86** $\pm 0.04$ |

from 2.92 to 2.71 for the unconditional setting. The experiment shows that Diff-Instruct is able to inject the knowledge of pre-trained diffusion models to enhance the generators further.

The results demonstrate that Diff-Instruct is a powerful method capable of improving existing GAN models that are supposed to converge with adversarial training. There are two possible reasons for such improvements. First, our Diff-Instruct utilizes well-trained diffusion models to supervise the generator. For instance, on the CIFAR10 dataset with conditional labels, the teacher EDM model can achieve the FID of $1.79$, which is significantly better than StyleGAN2 with an FID of $2.42$. Second, Diff-Instruct takes diffused data into account when minimizing the IKL, overcoming the potential degeneration issues of the divergences that adversarial training intends to minimize.

## 5  Discussion

This work presents a novel learning paradigm, Diff-Instruct, which is to our best knowledge, the first method that enables knowledge transfer from pre-trained diffusion models into generic generators in a data-free manner. The theoretical foundations and practical methods introduced in this work hold promise for advancing the utilization of diffusion generative models and implicit models across various domains and applications.

Nonetheless, Diff-Instruct has its limitations that call for further research along this line. First, with the abundance of powerful pre-trained DMs with diverse expertise, levering multiple models as instructors is another promising direction that is not investigated in this work. Second, even though data-free is a feature of our method, utilizing real data can potentially boost the learning process. The potential benefits of incorporating both Diff-Instruct and training data have not been explored yet. Lastly, in the extreme case where we have only data and no pre-trained DMs, our DI framework can still be adapted. Potentially we can train a teacher diffusion model with data, and concurrently, use it to instruct the student model. This indirect way of training may enable other generative models such as GANs, to enjoy the benefits of diffusion models, e.g., ease of training, ability to scale, and high sample quality, etc.

### Acknowledgments and Disclosure of Funding

W. Luo and Z. Zhang have been supported by the Beijing Natural Science Foundation (Z190001) and the National Natural Science Foundation of China (No. 12271011).

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

# A Technical details

## A.1 Robustness of Integral KL divergence

One of the benefits of using the Integral KL divergence over the traditional KL divergence is its robustness to misaligned density support. To illustrate this advantage, we consider a well-known example in [2]. Let $z$ be a random variable following the uniform distribution on the unit interval $[0,1]$. Consider $\mathbb{P}_0$ to be the distribution of $(0,z) \in \mathbb{R}^2$. Now let $g_\theta(z) = (\theta, z)$, where $\theta$ is a single real parameter. The density of $\mathbb{P}_0$ and $\mathbb{P}_\theta$ are $p_0(x, z) = \mathbb{I}_{x=0}(x)\mathbb{I}_{[0,1]}(z)$ and $p_\theta(x, z) = \mathbb{I}_\theta(x)\mathbb{I}_{[0,1]}(z)$. Since for each $\theta \neq 0$, the support of $p_\theta$ and $p_0$ does not intersect, the KL divergence between $\mathbb{P}_\theta$ and $\mathbb{P}_0$ is ill-defined with

$$\mathcal{D}_{KL}(\mathbb{P}_\theta, \mathbb{P}_0) = \begin{cases} +\infty & \theta \neq 0; \\ 0 & \theta = 0. \end{cases} \tag{A.1}$$

The same is also true for the Jensen-Shannon divergence where

$$\mathcal{D}_{JS}(\mathbb{P}_\theta, \mathbb{P}_0) = \begin{cases} \log 2 & \theta \neq 0; \\ 0 & \theta = 0. \end{cases} \tag{A.2}$$

So minimizing the KL divergence with a gradient-based algorithm does not lead the generator to converge to the correct parameter $\theta = 0$. However, IKL provides a finite and reliable objective for training the generator. More precisely, considering a simple diffusion

$$\mathrm{d}\boldsymbol{x}_t = \mathrm{d}\boldsymbol{w}_t. \tag{A.3}$$

The marginal distribution of $(x, z)$ under diffusion (A.3) initialized with $\mathbb{P}_0$ and $\mathbb{P}_\theta$ writes

$$p_0^{(t)}(x, z) = \mathcal{N}(x; 0, t) \int_0^1 \mathcal{N}(z; s, t)\mathrm{d}s,$$

$$p_\theta^{(t)}(x, z) = \mathcal{N}(x; \theta, t) \int_0^1 \mathcal{N}(z; s, t)\mathrm{d}s,$$

which are defined on $(x, z) \in \mathbb{R}^2$. The notation $\mathcal{N}(x; \mu, \sigma^2)$ represents the density function of Gaussian distribution with mean $\mu$ and variance $\sigma^2$. The IKL divergence with weight function $w(t)$ thus has the expression

$$\begin{aligned}
\mathcal{D}_{IKL}(\mathbb{P}_\theta, \mathbb{P}_0) &= \int_{t=0}^\infty w(t) \left[ \mathbb{E}_{(x,z) \sim p_\theta^{(t)}(x,z)} \log \frac{p_\theta^{(t)}(x,z)}{p_\theta^{(0)}(x,z)} \right] \mathrm{d}t \\
&= \int_{t=0}^\infty w(t) \left[ \mathbb{E}_{(x,z) \sim p_\theta^{(t)}(x,z)} \log \frac{\mathcal{N}(x; \theta, t)}{\mathcal{N}(x; 0, t)} \right] \mathrm{d}t \\
&= \int_{t=0}^\infty w(t) \left[ \mathbb{E}_{x \sim \mathcal{N}(x; \theta, t)} \log \frac{\mathcal{N}(x; \theta, t)}{\mathcal{N}(x; 0, t)} \right] \mathrm{d}t \\
&= \int_{t=0}^\infty w(t) \left[ \mathbb{E}_{x \sim \mathcal{N}(x; \theta, t)} \frac{1}{2t} \left[ x^2 - (x - \theta)^2 \right] \right] \mathrm{d}t \\
&= \int_{t=0}^\infty w(t) \left[ \mathbb{E}_{x \sim \mathcal{N}(x; \theta, t)} \frac{1}{2t} \left[ 2\theta x - \theta^2 \right] \right] \mathrm{d}t \\
&= \theta^2 \left[ \int_0^\infty \frac{w(t)}{2t} \mathrm{d}t \right], \theta \in \mathbb{R}.
\end{aligned} \tag{A.4}$$

By properly choosing the weighting function $w(t)$, $\mathcal{D}_{IKL}$ is finite as long as $\int_0^T \frac{w(t)}{2t}\mathrm{d}t$ is finite. For instance, $w(t) = 1/t$ if $t \geq 1$ and $w(t) = t$ if $t \leq 1$ as a simple choice [4]. The IKL divergence in (A.4) is a differentiable quadratic function of parameter $\theta$ with a single minima $\theta = 0$ which lead to the $\mathcal{D}_{IKL}(\mathbb{P}_\theta, \mathbb{P}_0) = 0$.

Table 5 shows a summary of the comparison among IKL divergence, KL divergence and the Wasserstein distance between $\mathbb{P}_\theta$ and $\mathbb{P}_0$. Our IKL is more suitable for learning $\theta$ with gradient-based optimization algorithms.

---

[4]In practice when distilling from pre-trained diffusion models, we use the same weighting function for training pre-trained diffusion models for Diff-Instruct, which is also inverted U-shaped.

Table 5: Comparison of divergence property against IKL.

| Divergence | distance | smooth |
|---|---|---|
| **IKL (ours)** | $\propto \theta^2$ | ✓ |
| KL | $+\infty$(A.1) | ✗ |
| Wasserstein | $\propto \lvert \theta \rvert$ [2] | ✗ |
| Jensen-Shannon | $\log 2$ (A.2) | ✗ |

## A.2 Proof of Theorem 3.3

*Proof.* Recall the definition of $q^{(t)}$, the sample is obtained by $\boldsymbol{x}_0 = g_\theta(\boldsymbol{z})$, $\boldsymbol{z} \sim p_z$, and $\boldsymbol{x}_t \vert \boldsymbol{x}_0 \sim p_t(\boldsymbol{x}_t \vert \boldsymbol{x}_0)$ according to forward SDE (2.1). Since the solution of forward, SDE is uniquely determined by the initial point $\boldsymbol{x}_0$ and a trajectory of Wiener process $\boldsymbol{w}_{t \in [0,T]}$, we slightly abuse the notation and let $\boldsymbol{x}_t = \mathcal{F}(g_\theta(\boldsymbol{z}), \boldsymbol{w})$ to represent the solution of $\boldsymbol{x}_t$ generated by $\boldsymbol{x}_0$ and $\boldsymbol{w}$. We let $\boldsymbol{w}_{[0,1]} \sim \mathbb{P}_{\boldsymbol{w}}$ to demonstrate a trajectory from the Wiener process where $\mathbb{P}_{\boldsymbol{w}}$ represents the path measure of Weiner process on $t \in [0,T]$. There are two terms that contain the generator's parameter $\theta$. The term $\boldsymbol{x}_t$ contains parameter through $\boldsymbol{x}_0 = g_\theta(\boldsymbol{z}), \boldsymbol{z} \sim p_z$. The marginal density $q^{(t)}$ also contains parameter $\theta$ implicitly since $q^{(t)}$ is initialized with $q^{(0)}$ which is generated by the generator. To demonstrate the parameter dependence, we may use $q_\theta^{(t)}$ to represent $q^{(t)}$.

The $p^{(t)}$ is defined through the pre-trained diffusion models with score functions $\boldsymbol{s}_{p^{(t)}}$. The IKL divergence between $q^{(t)}$ and $p^{(t)}$ is defined with,

$$
\begin{aligned}
\mathcal{D}_{IKL}^{[0,T]}(q_\theta^{(0)}, p^{(0)}) &:= \int_{t=0}^T w(t) \mathcal{D}_{KL}(q_\theta^{(t)}, p^{(t)}) \mathrm{d}t \\
&= \int_{t=0}^T w(t) \mathbb{E}_{\boldsymbol{x}_t \sim q_\theta^{(t)}} \Big[ \log \frac{q_\theta^{(t)}(\boldsymbol{x}_t)}{p^{(t)}(\boldsymbol{x}_t)} \Big] \mathrm{d}t \\
&= \int_{t=0}^T w(t) \mathbb{E}_{\substack{\boldsymbol{z} \sim p_z, \\ \boldsymbol{w} \sim \mathbb{P}_{\boldsymbol{w}}}} \Big[ \log \frac{q_\theta^{(t)}(\mathcal{F}(g_\theta(\boldsymbol{z}), \boldsymbol{w}))}{p^{(t)}(\mathcal{F}(g_\theta(\boldsymbol{z}), \boldsymbol{w}))} \Big] \mathrm{d}t
\end{aligned}
\tag{A.5}
$$

Taking the $\theta$ gradient of IKL (A.5), we have

$$
\begin{aligned}
&\frac{\partial}{\partial \theta} \mathcal{D}_{IKL}^{[0,T]}(q_\theta^{(t)}, p^{(t)}) \\
&= \frac{\partial}{\partial \theta} \int_{t=0}^T w(t) \mathbb{E}_{\substack{\boldsymbol{z} \sim p_z, \\ \boldsymbol{w} \sim \mathbb{P}_{\boldsymbol{w}}}} \Big[ \log \frac{q_\theta^{(t)}(\mathcal{F}(g_\theta(\boldsymbol{z}), \boldsymbol{w}))}{p^{(t)}(\mathcal{F}(g_\theta(\boldsymbol{z}), \boldsymbol{w}))} \Big] \\
&= \int_{t=0}^T w(t) \mathbb{E}_{\substack{\boldsymbol{z} \sim p_z, \\ \boldsymbol{w} \sim \mathbb{P}_{\boldsymbol{w}}}} \frac{\partial}{\partial \theta} \Big[ \log \frac{q_\theta^{(t)}(\mathcal{F}(g_\theta(\boldsymbol{z}), \boldsymbol{w}))}{p^{(t)}(\mathcal{F}(g_\theta(\boldsymbol{z}), \boldsymbol{w}))} \Big] \\
&= \int_{t=0}^T w(t) \mathbb{E}_{\substack{\boldsymbol{z} \sim p_z, \\ \boldsymbol{w} \sim \mathbb{P}_{\boldsymbol{w}}}} \nabla_{\boldsymbol{x}_t} \Big[ \log \frac{q_\theta^{(t)}(\mathcal{F}(g_\theta(\boldsymbol{z}), \boldsymbol{w}))}{p^{(t)}(\mathcal{F}(g_\theta(\boldsymbol{z}), \boldsymbol{w}))} \Big] \frac{\partial \mathcal{F}(g_\theta(\boldsymbol{z}), \boldsymbol{w})}{\partial \theta} \\
&\quad + \int_{t=0}^T w(t) \mathbb{E}_{\substack{\boldsymbol{z} \sim p_z, \\ \boldsymbol{w} \sim \mathbb{P}_{\boldsymbol{w}}}} \frac{\partial}{\partial \theta} \log q_\theta^{(t)}(\boldsymbol{x}_t) \big|_{\boldsymbol{x}_t = \mathcal{F}(g_\theta(\boldsymbol{z}), \boldsymbol{w})} \\
&= \int_{t=0}^T w(t) \mathbb{E}_{\substack{\boldsymbol{z} \sim p_z, \boldsymbol{w} \sim \mathbb{P}_{\boldsymbol{w}}, \boldsymbol{x}_0 = g_\theta(\boldsymbol{z}) \\ \boldsymbol{x}_t = \mathcal{F}(\boldsymbol{x}_0, \boldsymbol{w})}} \nabla_{\boldsymbol{x}_t} \Big[ \log \frac{q_\theta^{(t)}(\boldsymbol{x}_t)}{p^{(t)}(\boldsymbol{x}_t)} \Big] \frac{\partial \boldsymbol{x}_t}{\partial \theta} + \int_{t=0}^T w(t) \mathbb{E}_{\boldsymbol{x}_t \sim p_\theta^{(t)}} \frac{\partial}{\partial \theta} \log q_\theta^{(t)}(\boldsymbol{x}_t) \\
&= A + B.
\end{aligned}
\tag{A.6}
$$

The term $A$ in equation (A.6) writes

$$
A = \int_{t=0}^T w(t) \mathbb{E}_{\boldsymbol{x}_t \sim q^{(t)}} \Big[ \boldsymbol{s}_{q^{(t)}}(\boldsymbol{x}_t) - \boldsymbol{s}_{p^{(t)}}(\boldsymbol{x}_t) \Big] \frac{\partial \boldsymbol{x}_t}{\partial \theta}
\tag{A.7}
$$

We show that the term $B$ in equation (A.6) vanishes.

$$B = \int_{t=0}^{T} w(t)\mathbb{E}_{\boldsymbol{x}_t \sim q_\theta^{(t)}} \frac{\partial}{\partial\theta}\log q_\theta^{(t)}(\boldsymbol{x}_t)$$

$$= \int_{t=0}^{T} w(t)\int \frac{1}{p_\theta^{(t)}(\boldsymbol{x}_t)}\frac{\partial}{\partial\theta}q_\theta^{(t)}(\boldsymbol{x}_t)q_\theta^{(t)}(\boldsymbol{x}_t)\mathrm{d}\boldsymbol{x}_t$$

$$= \int_{t=0}^{T} w(t)\int \frac{\partial}{\partial\theta}q_\theta^{(t)}(\boldsymbol{x}_t)\mathrm{d}\boldsymbol{x}_t$$

$$= \int_{t=0}^{T} w(t)\frac{\partial}{\partial\theta}\int q_\theta^{(t)}(\boldsymbol{x}_t)\mathrm{d}\boldsymbol{x}_t \qquad (A.8)$$

$$= \int_{t=0}^{T} w(t)\frac{\partial}{\partial\theta}\mathbf{1}\mathrm{d}\boldsymbol{x}_t$$

$$= \mathbf{0}$$

$$(A.9)$$

The equality (A.8) holds if function $q_\theta^{(t)}(\boldsymbol{x})$ satisfies the conditions (1). $q_\theta^{(t)}(\boldsymbol{x})$ is Lebesgue integrable for $\boldsymbol{x}$ with each $\theta$; (2). For almost all $\boldsymbol{x} \in \mathbb{R}^D$, the partial derivative $\partial q_\theta^{(t)}(\boldsymbol{x})/\partial\theta$ exists for all $\theta \in \Theta$. (3) there exists an integrable function $h(.): \mathbb{R}^D \to \mathbb{R}$, such that $q_\theta^{(t)}(\boldsymbol{x}) \leq h(\boldsymbol{x})$ for all $\boldsymbol{x}$ in its domain. Then the derivative w.r.t $\theta$ can be exchanged with the integral over $\boldsymbol{x}$, i.e.

$$\int \frac{\partial}{\partial\theta}q_\theta^{(t)}(\boldsymbol{x})\mathrm{d}\boldsymbol{x} = \frac{\partial}{\partial\theta}\int q_\theta^{(t)}(\boldsymbol{x})\mathrm{d}\boldsymbol{x}.$$

$\square$

**Remark A.1.** In practice, most commonly used forward diffusion processes can be expressed as a form of scale and noise addition:

$$\boldsymbol{x}_t = \alpha(t)\boldsymbol{x}_0 + \beta(t)\epsilon, \quad \epsilon \sim \mathcal{N}(\epsilon; \mathbf{0}, \mathbf{I}). \qquad (A.10)$$

So the term $\boldsymbol{z} \sim p_z$, $\boldsymbol{w} \sim \mathbb{P}_{\boldsymbol{w}}$, $\boldsymbol{x}_t = \mathcal{F}(\boldsymbol{x}_0, \boldsymbol{w})$ in equation (A.5) can be instantiated as $\boldsymbol{z} \sim p_z, \epsilon \sim \mathcal{N}(\epsilon; \mathbf{0}, \boldsymbol{I}), \boldsymbol{x}_t = \alpha(t)\boldsymbol{x}_0 + \beta(t)\epsilon$.

### A.3 Proof of Corollary 3.4

*Proof.* Since the $q(\boldsymbol{x}_0) = \delta_{g(\theta)}(\boldsymbol{x}_0)$, thus the conditional distribution and marginal distribution coincides, i.e.

$$q^{(t)}(\boldsymbol{x}_t) = \int p_t(\boldsymbol{x}_t|\boldsymbol{x}_0)q^{(0)}(\boldsymbol{x}_0)\mathrm{d}\boldsymbol{x}_0 = p_t(\boldsymbol{x}_t|\boldsymbol{x}_0)\mathbb{I}_{g(\theta)}(\boldsymbol{x}_0) = p_t(\boldsymbol{x}_t|g(\theta))$$

So the marginal score function writes

$$\boldsymbol{s}_{q^{(t)}}(\boldsymbol{x}_t) := \nabla_{\boldsymbol{x}_t}\log q^{(t)}(\boldsymbol{x}_t) = \nabla_{\boldsymbol{x}_t}\log p_t(\boldsymbol{x}_t|g(\theta))$$

So the gradient formula (3.2) turns to

$$\mathrm{Grad}(\theta) = \int_{t=0}^{T} w(t)\mathbb{E}_{\substack{\boldsymbol{x}_0 = g(\theta), \\ \boldsymbol{x}_t|\boldsymbol{x}_0 \sim p_t(\boldsymbol{x}_t|\boldsymbol{x}_0)}} \left[\nabla_{\boldsymbol{x}_t}\log p_t(\boldsymbol{x}_t|\boldsymbol{x}_0) - \boldsymbol{s}_{p^{(t)}}(\boldsymbol{x}_t)\right]\frac{\partial\boldsymbol{x}_t}{\partial\theta}\mathrm{d}t.$$

$\square$

If we define a $\epsilon$-network $\epsilon_p(x, t) := -\boldsymbol{s}_{p^{(t)}}(\boldsymbol{x}_t)/\sigma(t)$ and consider the forward diffusion $p_t(\boldsymbol{x}_t|\boldsymbol{x}_0) = \mathcal{N}(\boldsymbol{x}_t; \alpha(t)\boldsymbol{x}_0, \sigma(t)\boldsymbol{I})$, the forward diffusion can be implemented with $\boldsymbol{x}_t = \alpha(t)\boldsymbol{x}_0 + \sigma(t)\epsilon, \epsilon \sim \mathcal{N}(\epsilon; \mathbf{0}, \boldsymbol{I})$, then the objective turns to

$$\mathrm{Grad}^{(SDS)}(\theta) = \int_0^T \frac{w(t)}{\sigma(t)}\mathbb{E}_{\substack{\boldsymbol{x}_0 = g_\theta(\boldsymbol{z}_0), \epsilon \sim \mathcal{N}(\mathbf{0}, \mathbf{I}), \\ \boldsymbol{x}_t = \alpha(t)\boldsymbol{x}_0 + \sigma(t)\epsilon}}\left[\epsilon_p(\boldsymbol{x}_t, t) - \epsilon\right]\frac{\partial\boldsymbol{x}_t}{\partial\theta}\mathrm{d}t, \qquad (A.11)$$

which recovers the SDS gradient estimation proposed in DreamFusion [54]. In summary, we have shown that our Diff-Instruct's gradient is equivalent to the gradient formula of score distillation sampling (SDS) when the generator outputs a single data that is differentiable to the generator's parameters. This is not a coincidence, as a NeRF model can be viewed as a generator that outputs a Dirac's Delta distribution when the view direction is fixed. Therefore, using SDS to learn a text-conditioned NeRF model is essentially an application of using an approximated version of Diff-Instruct to distill a pre-trained text-to-2D diffusion model in order to obtain a NeRF.

## A.4 Proof of Corollary 3.5

*Proof.* Following the same notation as in section 2, consider the adversarial training that minimizes the KL divergence as we introduced in Section 2. The learning objective is

$$\mathcal{L}^{(KL)}(\theta) = \mathbb{E}_{\boldsymbol{z} \sim p_z}[\log \frac{1 - h(g_\theta(\boldsymbol{z}))}{h(g_\theta(\boldsymbol{z}))}] \tag{A.12}$$

Assume the discriminator $h(.)$ is optimal, then

$$h(\boldsymbol{x}) = \frac{p_d(\boldsymbol{x})}{p_d(\boldsymbol{x}) + p_g(\boldsymbol{x})}, and \log \frac{1 - h(\boldsymbol{x})}{h(\boldsymbol{x})} = \log \frac{p_d(\boldsymbol{x})}{p_g(\boldsymbol{x})}$$

Then gradient for the generator parameter $\theta$ turn to

$$\frac{\partial}{\partial \theta} \mathcal{L}^{(KL)}(\theta) = \mathbb{E}_{\boldsymbol{z} \sim p_z} \nabla_{\boldsymbol{x}} (\log \frac{1 - h(\boldsymbol{x})}{h(\boldsymbol{x})})|_{\boldsymbol{x} = g_\theta(\boldsymbol{z})} \frac{\partial g_\theta(\boldsymbol{z})}{\partial \theta}$$

$$= \mathbb{E}_{\boldsymbol{z} \sim p_z} \nabla_{\boldsymbol{x}} (\log \frac{p_d(\boldsymbol{x})}{p_g(\boldsymbol{x})})|_{\boldsymbol{x} = g_\theta(\boldsymbol{z})} \frac{\partial g_\theta(\boldsymbol{z})}{\partial \theta}$$

$$= \mathbb{E}_{\boldsymbol{z} \sim p_z} \Big[ \nabla_{\boldsymbol{x}} \log p_d(\boldsymbol{x}) - \nabla_{\boldsymbol{x}} \log p_g(\boldsymbol{x}) \Big] |_{\boldsymbol{x} = g_\theta(\boldsymbol{z})} \frac{\partial g_\theta(\boldsymbol{z})}{\partial \theta}$$

$$= \mathbb{E}_{\substack{\boldsymbol{z} \sim p_z, \\ \boldsymbol{x} = g_\theta(\boldsymbol{z})}} \Big[ \nabla_{\boldsymbol{x}} \log p_d(\boldsymbol{x}) - \nabla_{\boldsymbol{x}} \log p_g(\boldsymbol{x}) \Big] \frac{\partial \boldsymbol{x}}{\partial \theta}$$

$$= \mathbb{E}_{\substack{\boldsymbol{z} \sim p_z, \\ \boldsymbol{x} = g_\theta(\boldsymbol{z})}} \Big[ \boldsymbol{s}_d(\boldsymbol{x}) - \boldsymbol{s}_g(\boldsymbol{x}) \Big] \frac{\partial \boldsymbol{x}}{\partial \theta} \tag{A.13}$$

where $\boldsymbol{s}_g(\boldsymbol{x}) = \nabla_{\boldsymbol{x}} \log p_g(\boldsymbol{x})$ and $\boldsymbol{s}_d(\boldsymbol{x}) = \nabla_{\boldsymbol{x}} \log p_d(\boldsymbol{x})$ denote the score function of the generator and the data distribution. The equation (A.13) shows that the adversarial training that minimizes the KL divergence is equivalent to the gradient formula (3.2) with a special weight function $w(0) = 1$ and $w(t) > 0$ for all $t > 0$, if the discriminator can be trained to be optimal. □

### A.4.1 Comparison of Distillation Methods

Table 6: Comparison of Diffusion Distillation Methods. The term *efficiency* represents the training efficiency of DMs. *flexibility* means whether the student model needs to have the same in-out dimensions.

| Method | data-free | flexibility | efficiency |
|---|---|---|---|
| ReFlow[40] | ✓ | ✗ | *low* |
| DFNO[82] | ✓ | ✗ | *low* |
| KD[42] | ✓ | ✗ | *low* |
| CD[66] | ✗ | ✗ | *medium* |
| PD[61] | ✓ | ✗ | *low* |
| **DI** (ours) | ✓ | ✓ | *high* |

## B   More on experiments

To demonstrate the efficacy of our proposed Diff-Instruct, we choose the state-of-the-art EDMs [32] as instructors, which have achieved state-of-the-art generative performance on several benchmarks

Table 7: Hyperparameters used for Diff-Instruct for Diffusion Distillation

| Hyperparameter | CIFAR-10 (Uncond) | | ImageNet $64 \times 64$ | | CIFAR-10 (Cond) | |
|---|---|---|---|---|---|---|
| | DM $s_\phi$ | Generator $g_\theta$ | DM $s_\phi$ | Generator $g_\theta$ | DM $s_\phi$ | Generator $g_\theta$ |
| Learning rate | 1e-5 | 1e-5 | 1e-5 | 1e-5 | 1e-5 | 1e-5 |
| Batch size | 64 | 64 | 96 | 96 | 64 | 64 |
| $\sigma(t^*)$ | 2.5 | 2.5 | 5.0 | 5.0 | 2.5 | 2.5 |
| $Adam\ \beta_0$ | 0.0 | 0.0 | 0.0 | 0.0 | 0.0 | 0.0 |
| $Adam\ \beta_1$ | 0.99 | 0.99 | 0.99 | 0.99 | 0.99 | 0.99 |
| Training iterations | 100k | 100k | 50k | 50k | 100k | 100k |
| Number of GPUs | 4 | 4 | 8 | 8 | 4 | 4 |

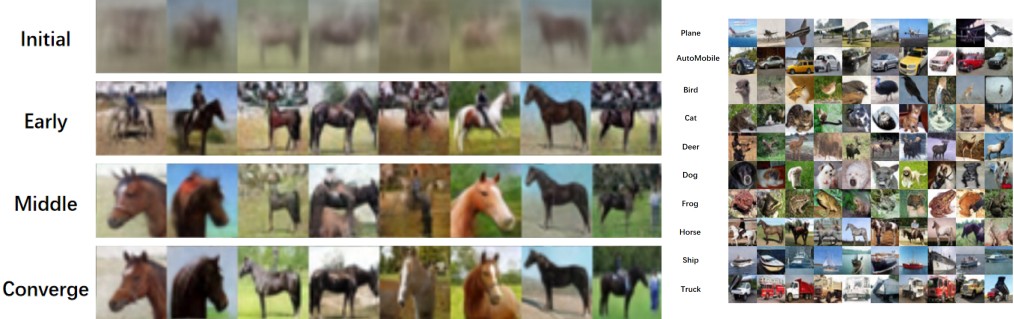

Figure 4: *Left*: Demonstration of the training process for Diff-Instruct for diffusion distillation. Generated samples with the same latent vectors and different generator weights during training are put from the top to the bottom. *Right*: generated samples from a one-step generative model distilled from pre-trained class-conditional EMD model on the CIFAR10 dataset.

such as CIFAR10 and ImageNet $64 \times 64$. The EDM model depends on the diffusion process

$$d\boldsymbol{x}_t = G(t)d\boldsymbol{w}_t, t \in [0, T]. \tag{B.1}$$

Samples from the forward process (B.1) can be generated by adding random noise to the output of the generator function, i.e., $\boldsymbol{x}_t = \boldsymbol{x}_0 + \sigma(t)\boldsymbol{\epsilon}$ where $\boldsymbol{\epsilon} \sim \mathcal{N}(\boldsymbol{0}, \boldsymbol{I})$ is a Gaussian vector and $\sigma(t) := \sqrt{\int_0^t G^2(s)ds}$ is a function with explicit expressions. We download the pre-trained model checkpoints from the official website[5] and consider transferring their knowledge to implicit generative models, specifically UNet and StyleGAN as generators.

We calculate the FID and IS in the same way as the StyleGAN2-ADA[6] codebase. For the ImageNet $64 \times 64$ dataset, we use the same pre-processing scripts as the EDM model on ImageNet $64 \times 64$ dataset.

### B.1 Detailed experimental settings of diffusion distillation

When the downstream generative model is a UNet generator, our DI provides another way for diffusion distillation, directly competing with progressive distillation [61] or consistency distillation [66].

We initialize the diffusion model for the implicit distribution $s_\phi$ with the weight parameters of pre-trained DM. The initialization of the generator is put in the following paragraph. With the pre-trained DM, initialized generator, and the initialized DM for implicit distribution, we use our Diff-Instruct algorithm 1 to update the DM for implicit distribution and the generator's parameters. We use the Adam optimizer for both the DM and the generator. For the generator, we use the same exponential moving average (EMA) technique as the EDM model's training scripts.

To make a fair comparison, we do not compare approaches that involve other components such as classifier guidance or additional architectures for the generation as [82] because these models have significantly larger model sizes for inference.

---

[5]https://github.com/NVlabs/edm
[6]https://github.com/NVlabs/stylegan2-ada-pytorch

Table 8: Comparison of Memory Costs and Wall-Clock Timing of CD and Diff-Instruct. The Peak GPU-Memo shows the Maximum observed GPU memory caused by the Tensor and Computational Graph of the host GPU. Sec-per-K Iterations report the wall-clock time for each training iteration. For Diff-Instruct, each iteration consists of two stages as in Algorithm 1 in the main text. Test environment: PyTorch 1.12.1 and Torchvision 0.13.1, and Torch.distributed.parallel on 2 V100 GPUs.

| Method | Peak GPU-Memo(GB) | Peak CPU-Memo(GB) | Sec-per-K Iterations |
|---|---|---|---|
| CD | 9.55 | 2.75 | 0.0489 |
| Diff-Instruct | 10.40 | 2.78 | 0.0728 |

**Initialization of the generator.** When the generator is chosen to have the same architecture as the pre-trained diffusion models (i.e. UNet in most cases), we can initialize the generator with a special method with the score network of the pre-trained DM. Taking the forward diffusion process (B.1) as an instance. The time-indexed score functions explicitly define a data-prediction transform through Tweedie's formula

$$\widehat{x}_0 := x_t + \sigma(t)^2 \nabla_{x_t} \log q^{(t)}(x_t) \tag{B.2}$$

This formula tells that the marginal score functions can be converted to a data-prediction transform which can transform noisy data to clean data. Motivated by this property, we initialize the implicit generator $g_\theta$ for Diff-Instruct in Algorithm 1 via a score network-induced data-prediction transform of the teacher diffusion models at some fixed time $t^*$:

$$x = z + \sigma^2(t^*) s_{p^{(t^*)}}(z). \tag{B.3}$$

where $z \sim \mathcal{N}(z; 0, \sigma^2(t^*)I)$ is the latent vector. This generator takes a Gaussian noise with the variance $\sigma(t^*)$ and zero mean as an input latent vector.

We find that each noise level $\sigma(t^*)$ can give a comparable initialization, so we roughly selected an $\sigma(t^*)$ for different datasets. We put detailed hyper-parameters for distilling in Table 7. The left hand of Figure 4 gives a demonstration of the samples with the same latent vectors from different generators during training. The top line is the initialized generator (i.e. modified from pre-trained diffusion models). During the training, the initialized generator is trained to generate high-quality samples. The right hand of Figure 4 shows class-conditioned samples from a one-step model (generator) distilled from a class-conditional EDM model on the CIFAR10 dataset.

**Comparison of Computational Costs** The Diff-Instruct algorithm involves training an additional auxiliary diffusion model during the training phases. It is worth noting that although this auxiliary diffusion model brings additional memory cost, this additional memory cost is very limited because the memory bottleneck of training lies in the computational graph of the backpropagation, instead of only saving one more model.

In the Diff-Instruct algorithm, the model and the generator are updated alternatively, which means that the other model's parameters are fixed and do not participate in back-propagation when one model is being updated. So the memory cost for back-propagating through the computational graph is almost the same as one model.

To quantitatively measure how much additional computational costs are brought in, we run an experiment to compare the computational and memory costs of Diff-Instruct with our baseline method, the Consistency Distillation, in 8. The test was run on 2 Nvidia V100 GPUs with 128 batch size and PyTorch distributed data-parallel mechanism.

The result shows that Diff-Instruct brings in minor additional memory costs than CD (10.40 over 9.55). This is because the Diff-Instruct only needs additional GPU memory to save the auxiliary model $s_\phi$. But the $s_\phi$ and generator $g_\theta$ are updated alternatively, so their computational graph does not interact. As a result, the memory bottleneck caused by computational graph and back-propagation does not bring more costs to Diff-Instruct.

As for the wall-clock time for 1K iterations, we see that Diff-Instruct costs 0.0728 seconds, while the CD costs 0.0489 seconds. This is because each iteration of Diff-Instruct consists of two alternate steps as we show in Algorithm 1. Overall, the Diff-Instruct costs almost the same GPU and CPU memory as the baseline CD, but about 1.5 times wall-clock time than the CD for each iteration.

Table 9: Hyperparameters used for Diff-Instruct for GAN Improvement on CIFAR10 dataset under different settings.

| Hyperparameter | StyleGAN-2 + Cond | | StyleGAN-2 + Uncond | | StyleGAN-2-ADA + Cond | | StyleGAN-2-ADA + Uncond | |
| --- | --- | --- | --- | --- | --- | --- | --- | --- |
| | DM $s_\phi$ | Generator $g_\theta$ | DM $s_\phi$ | Generator $g_\theta$ | DM $s_\phi$ | Generator $g_\theta$ | DM $s_\phi$ | Generator $g_\theta$ |
| Learning rate | 2e-7 | 2e-7 | 1.5e-6 | 1.5e-6 | 1.5e-6 | 1.5e-6 | 1.5e-6 | 1.5e-6 |
| Batch size | 512 | 512 | 512 | 512 | 512 | 512 | 512 | 512 |
| $Adam\ \beta_0$ | 0.0 | 0.0 | 0.0 | 0.0 | 0.0 | 0.0 | 0.0 | 0.0 |
| $Adam\ \beta_1$ | 0.99 | 0.99 | 0.99 | 0.99 | 0.99 | 0.99 | 0.99 | 0.99 |
| Training iterations | 16k | 16k | 6k | 6k | 9k | 9k | 10k | 10k |
| Number of GPUs | 4 | 4 | 4 | 4 | 4 | 4 | 4 | 4 |

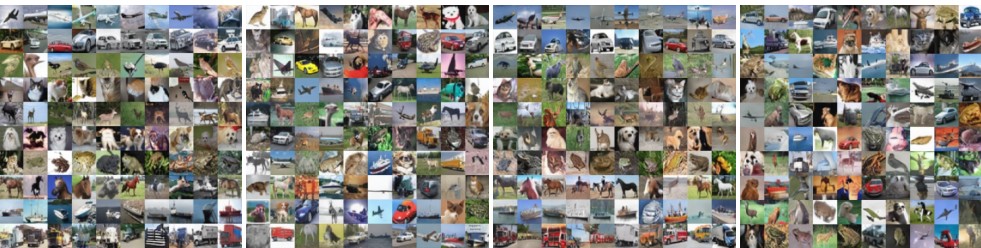

Figure 5: Generated samples from improved generators with Diff-Instruct in Table 3 and 1. The FID for generated samples from left to right is 2.27, 2.71, 6.62, and 7.56.

## B.2  Detailed experimental settings of GAN improvement

This experiment aims to show the power of Diff-Instruct to improve GAN's generator under different settings by transferring knowledge from pre-trained DMs.

**Experiment settings.**  The StyleGAN-2 model is a competitive GAN model on benchmarking datasets such as the CIFAR10 dataset, so we take StyleGAN-2 as the representative GAN model. We take the pre-trained EDM models with VP architecture [7] on the CIFAR10 datasets as pre-trained diffusion models.

We download the checkpoint of the pre-trained StyleGAN-2 models with Adaptive Data Augmentation (ADA)[29] from the official website[8]. We pre-train a StyleGAN-2 model [30] following the same configuration of the original paper. All models converge with adversarial training under different settings (conditional or unconditional with different data augmentation strategies). We initialize the generator with the weights of pre-trained GAN generators. We initialize the DM $s_\phi$ for implicit distribution with the same weights as the pre-trained DMs. We then use the Diff-Instruct with pre-trained DMs to improve the generator. For both the implicit DM $s_\phi$ and the generator $g_\theta$, we use the Adam optimizer to update the parameters. For the generator, we use the same exponential moving average technique as the official implementation of StyleGAN-ADA with Pytorch. We put detailed hyper-parameters for each experiment in Table 9.

---

[7] https://github.com/NVlabs/edm
[8] https://github.com/NVlabs/stylegan2-ada-pytorch

