# OpenReview forum: "Diff-Instruct: A Universal Approach for Transferring Knowledge From Pre-trained Diffusion Models"
_NeurIPS.cc/2023/Conference — NeurIPS 2023 poster_

### Official Review · Reviewer_kX6X · 2023-07-03

**Soundness:** 2 fair
**Presentation:** 1 poor
**Contribution:** 2 fair
**Rating:** 5
**Confidence:** 2

**Summary:**

The paper discusses the distillation of pre-trained diffusion models. The paper introduces a framework called Diff-Instruct, which allows for the transfer of knowledge from pre-trained DMs to other generative models in a data-free manner. Diff-Instruct utilizes a mathematical foundation based on minimizing a divergence called Integral Kullback-Leibler (IKL) divergence. This divergence is specifically designed for DMs and is more robust in comparing distributions with misaligned supports. The text also highlights connections to existing works such as DreamFusion and generative adversarial training. The effectiveness of Diff-Instruct is demonstrated through two scenarios: distilling pre-trained diffusion models and refining existing GAN models. Experimental results show that Diff-Instruct achieves state-of-the-art performance in single-step diffusion-based models and consistently improves pre-trained generators of GAN models across various settings.

**Strengths:**

* The paper tackles the problem of distilling diffusion models and addresses the important topic of accelerating inference for DMs.

* The proposed method utilizes a novel formulation based on the Integral Kullback-Leibler (IKL) divergence, which adds further interest to the approach. The authors make an effort trying to explain why this divergence is a reasonable choice for distillation.

* The experimental results showcased the effectiveness of this method, particularly in the context of 1-step distillation on datasets such as CIFAR and ImageNet64, where the results were highly promising.

These findings demonstrate the potential of the proposed method for enhancing the efficiency and performance of diffusion models in small domains.

**Weaknesses:**

* While the distillation results for CIFAR and ImageNet using the proposed method are good, they may not have practical utility in larger domains. Distillation becomes particularly crucial for such larger domains, and it remains unclear whether the method will be effective with larger images.

* Additionally, although the method doesn't require additional data, it does necessitate the use of two extra models, namely the student model and an auxiliary model for gradient propagation. This requirement can be prohibitive when dealing with larger domains and bigger models.

* I have some doubts about the notation (see questions). I would try to be consistent with the notation in the literature (see https://arxiv.org/abs/2011.13456).

**Questions:**

* When discussing knowledge transfer, what is the primary objective? Is it knowledge distillation or transfer? It would be helpful if you could simplify and specifically tie your contributions to the results of your experiments.

* Regarding Algorithm 1: when updating $\theta$, shouldn't $s$ be function of $\theta$? Could you clarify how the gradient has been formulated?

* In Equation 1, does $q$ represent the forward or reverse process in a Diffusion Model? It appears you're using $q$ for the reverse process of a pre-trained Diffusion Model, which is conventionally represented as the approximate posterior. Could you clarify this point?

* Furthermore, referencing Equation 1 in this paper (https://arxiv.org/abs/2011.13456), it seems that the notations in Equations 2.2 and 3.2 are inconsistent. Could you please provide some clarification?

* After training the implicit model, what role does $s_{\phi}$ serve? Is it merely an auxiliary model, and if so, how do you initialize it?

* How much time is required to distill a pre-trained diffusion model?

* You've mentioned that the Inverse Kullback-Leibler (IKL) divergence is more robust. Could you explain how you've arrived at this conclusion?

**Limitations:**

* Notation confusing or inconsistent.

* Not clear what is the goal of the paper (transfer vs distillation).

---

> ### Author Rebuttal · Authors · 2023-08-10
>
> Thank you for your useful feedback. We will address your concerns one by one in the following paragraphs.
> **Q1**. it remains unclear whether the method will be effective with larger images
>
> **A1**. We agree that larger-scale experiments, such as the text-to-image generation experiments are important to show the scaling ability of Diff-Instruct. However, due to the massive computational requirements and engineering implementations for large-scale generative modeling, we leave the scaling of Diff-Instruct to larger domains in future work.  However, in the rebuttal period, we try the Diff-Instruct on distilling Stable Diffusion into one step and got positive results. We put a visualization of a batch of images that are generated by a one-step generator distilled from Stable Diffusion in **Figure 1** on the additional page in the global author rebuttal cell. Our results indicate the scalability of Diff-Instruct to large-scale text-to-image diffusion models, and we plan to leave more exploration and engineering in our future work.
>
> **Q2**. two extra models and costs
>
> **A2**. We acknowledge that the additional diffusion model ($s_\phi$ in Algorithm 1) brings additional memory cost. However, this additional memory cost is limited because the memory bottleneck of training lies in the computational graph of the backpropagation, instead of only saving one more model. In Diff-Instruct, the model $s_\phi$ and the generator $g_\theta$ are updated alternatively, which means that the other model's parameters are fixed and do not participate in back-propagation when one model is being updated. So the memory cost for back-propagating through the computational graph is almost the same as one model. To quantitatively measure how much additional computational costs are brought in, we compare Diff-Instruct and CD's memory and computational costs in **Table 3** in the global author's rebuttal cell.
>
> The result show Diff-Instruct costs almost the same GPU and CPU memory as the baseline CD, but about 1.5 times wall-clock time than the CD for each iteration.
>
> Besides, our empirical evaluation also reveals that the Diff-Instruct converges significantly faster than the baseline method, the consistency distillation (CD). We record the FID curve w.r.t training iterations of Diff-Instruct and CD in the Left part of **Figure 2** on the additional page in the global author rebuttal cell. The results show that Diff-Instruct converges significantly faster than CD. For instance, the FID of Diff-Instruct converges to 4.6 with 7k training iterations, while the consistency distilled student model still holds an FID over 30.
>
> **Q4**. When discussing knowledge transfer, what is the primary objective?
>
> **A4**. We are sorry for the confusion of the word transferring and distillation. Since these two words do not have a clear definition in the literature, we abuse their use to mean to use a pre-trained diffusion model to guide the training of a one-step generator.
>
> **Q5**. Could you clarify how the gradient has been formulated?
>
> **A5**. In algorithm 1, when updating $\theta$, we fix the parameter of $s_\phi$ and use $s_\phi(.,t)$ and pre-trained score $s_{q^{(t)}}$ to compute the gradient of the parameter $\theta$. In the gradient formula, for a fixed time $t$, the term $x_0 = g_\theta(z)$ and $x_t|x_0\sim q_t(x_t|x_0)$ contain parameter $\theta$. Other components do not contain $\theta$. As for the implementation of the gradient, we random sample $x_0 = g_\theta(z)$ and $x_t|x_0\sim q_t(x_t|x_0)$, then we compute a coefficient of $coeff = w(t)[s_\phi(x_t) - s_{q^{(t)}}(x_t)].\operatorname{detach()}$. The $\operatorname{detach()}$ operator means to cut off the parameter dependency of the current node in the computation graph. Then we compute a loss function $\operatorname{loss}(\theta) = \langle coeff, x_t(\theta) \rangle$ and backward through the loss functions. So we have $\frac{\partial}{\partial\theta}\operatorname{L}(\theta) = w(t)[s_\phi(x_t) - s_{q^{(t)}}(x_t)]\frac{\partial x_t(\theta)}{\partial}$, which implements the gradient formula of Diff-Instruct at a certain time level.
>
> **Q3,6-7**. some doubts about the notation. does represent the forward or reverse process in a Diffusion Model?
>
> **A3,6-7**. In equations (2.2), (3.2), and Algorithm 1, We use the notion $q_t(x_t|x_0)$ to represent the forward process and the notion $q^{(t)}$ for the forward marginal distribution. This is different from Eq. (1) of the score sde paper which uses $p$ to represent the forward.
>
> **Q8**. After training the implicit model, what role does  serve? how do you initialize it?
>
> **A8**. After training, we discard the auxiliary model $s_\phi$ and only take the generator $g_\theta$ for inference. For all experiments, we initialize $s_\phi$ using the same architecture and weights of the pre-trained model $s_{q^{(t)}}$. This initialization works well across all experiments.
>
> **Q9**. How much time is required to distill a pre-trained diffusion model?
>
> **A9**. In practice, we find that using Diff-Instruct to train a one-step generator on CIFAR10 to converge takes less than 8 wall-clock hours on a single V100 GPU (with batch size 96), this shows that Diff-Instruct is quite efficient.
>
> **Q10**. Explain why IKL is more robust.
>
> **A10**. In Appendix A.1, we show that under a famous example with misaligned support from the Wasserstein GAN paper, the KL divergence is ill-defined (diverges to infinity)  due to misaligned density support. While the IKL is well-defined because it integrates the divergence after diffusion. This shows that IKL is more robust than KL divergence when measuring the distributions that have misaligned supports.
>
> Thank you for your kind suggestions. We hope our answers have resolved your concerns, and if you still have any concerns, please do let us know.

---

> > ### Comment · Reviewer_kX6X · 2023-08-13
> >
> > Thank you for the clarifications. I read the rebuttal and it answers my questions.

---

> > > ### Author Response · Authors · 2023-08-20
> > > **Thank you.**
> > >
> > > We are glad that we have resolved your concerns. Thank you for your questions and discussions.

---

### Official Review · Reviewer_6Ze7 · 2023-07-07

**Soundness:** 3 good
**Presentation:** 2 fair
**Contribution:** 2 fair
**Rating:** 6
**Confidence:** 3

**Summary:**

This work distills diffusion models to a GAN-style generator. The author provides mathmatic connection to other works, as well as experimental comparisons between the distilled GAN-style generator with other (mostly) diffusion models.


**Strengths:**

- This work introduce a way to distill diffusion model to GAN-style generator.
- And shows the connection to other works.
- The experiments show the proposed method works well.


**Weaknesses:**

- Lack baseline method. Table1 and Table2 shows the performance many diffusion methods, but they are either diffusion model trained on different datasets or distilled diffusion model that may not use the same teather model as the author used (i.e. EDM). In this case, the numbers in table1 &  2 are not compariable. There should be at least one baseline method that use the same teather model (EDM), the same generator architecture (one-step GAN-style generator), but different distillation method. Here is a simple baseline idea,  generating N images by the teather model (EDM), training a GAN on the generated images.

- Lack time comparison. One big benefit of one-step generator (as well as distillation) is inference speed. Otherwise, we can use the teacher model directly.





**Questions:**

Please check the weaknesses.

**Limitations:**

this work has no negative societal impact.

---

> ### Author Rebuttal · Authors · 2023-08-10
>
> Thank you for your reviews.  We will address your concerns one by one in the following paragraphs.
>
> **Q1**. Lack of baseline method.
>
> **A1**. We are sorry for the confusion. In Table 1 and Table 2 in the main text, the CD, PD, and Diff-Instruct use the same teacher model and the same student model. To make our representation more clear, we re-organize Table 1 and Table 2 to **Table1** and **Table2** in order to emphasize each teacher and student model.
>
> **Table1**. Unconditional sample quality on CIFAR10 through diffusion generations. ∗Methods that require synthetic data construction for distillation. †Methods that require real data for distillation.
> | Method                    | Teacher   | Student   | NFE       | FID       | IS        |
> | :------:                  | :------:  | :------:  | :------:  | :------:  | :------:  |
> | Pre-train + distillation  |
> | 1-ReFlow (+distill)∗      | ReFlow    | UNet      | 1         | 6.18      | 9.08      |
> | 2-ReFlow (+distill)∗      | ReFlow    | UNet      | 1         | 4.85      | 9.01      |
> | 3-ReFlow (+distill)∗      | ReFlow    | UNet      | 1         | 5.21      | 8.79      |
> | KD∗                       | DDPM      | UNet      | 1         | 9.36      |           |
> | CD-LPIPS†                 | EDM       | UNet      | 2         | 2.93      | 9.75      |
> | PD                        | EDM       | UNet      | 2         | 5.58      | 9.05      |
> | PD                        | EDM       | UNet      | 1         | 8.34      | 8.69      |
> | CD-L2†                    | EDM       | UNet      | 1         | 7.90      | 8.69      |
> | CD-LPIPS†                 | EDM       | UNet      | 1         | 3.55      | 9.48      |
> | Diff-Instruct             | EDM       | UNet      | 1         | 4.53      | 9.89      |
>
> **Table2**. ImageNet 64×64 through diffusion generations. ∗Methods that require synthetic data construction for distillation. †Methods that require real data for distillation.
> | Method                    | Teacher   | Student   | NFE       | FID       |
> | :------:                  | :------:  | :------:  | :------:  | :------:  |
> | Pre-train + distillation  |
> | GGDM∗                     | DDPM      | UNet      | 25        | 18.4      |
> | PD                        | EDM       | UNet      | 2         | 8.95      |
> | CD-LPIPS†                 | EDM       | UNet      | 2         | 4.70      |
> | PD                        | EDM       | UNet      | 1         | 15.39     |
> | CD-L2†                    | EDM       | UNet      | 1         | 12.10     |
> | CD-LPIPS†                 | EDM       | UNet      | 1         | 6.20      |
> | Diff-Instruct             | EDM       | UNet      | 1         | 5.57      |
>
> In particular, Table 1 and Table 2 in the main text show that the EDM model is the strongest among all diffusion models, so in our work, we use EDM models as the teacher models. In **Table 1** and **Table 2**, PD, CD, and Diff-Instruct all use the EDM teacher, and the same UNet student with the same architecture as the teacher model. (The author of consistency distillation re-implemented the PD for EDM, so the reported FID for PD is lower than that in PD's original paper)
>
> For the GAN improvement experiment, we compare the StyleGAN2 generator trained with and without Diff-Instruct in Tables 3 and 4. The StyleGAN2 generator is obtained from the official release. We use the pre-trained EDM as the teacher model for Diff-Instruct.
>
> **Q2**. Lack time comparison. One big benefit of one-step generator (as well as distillation) is inference speed. Otherwise, we can use the teacher model directly.
>
> **A2**. In Table 1 in the main text, the Diff-Instruct generator uses the same UNet architecture as the teacher diffusion model, so the number of sampling steps (NFE) represents its inference time costs. As we show in the upper part of Table 2 in the main text, sampling from our learned one-step generator with 1 NFE results in an FID of 4.19, which is significantly better than its teachers with 10 NFEs with an FID of 15.56. So we conclude that the one-step generator trained with Diff-Instruct achieves at least 10+ times acceleration (more efficient) than its teacher diffusion model.
>
> Thank you for your kind suggestions. We hope our answers have resolved your concerns, and if you still have any concerns, please let us know.

---

> > ### Comment · Reviewer_6Ze7 · 2023-08-14
> >
> > Thanks for you answers. Now table 1 and table 2 are better and clear.
> >
> > When comparing methods, it is better to fix other factors. For example, in table 1, it is hard to compare ReFlow (+distill)∗ and Diff-Instruct, because they use different Teachers (which are trained in different way and probabily on different data) and different data to "distill".  So in table 1 and table 2, we can safely compare Diff-Instruct with PD, but not others.
> >
> > Table 3 and table 4, Cifar10 conditional class sampling performance, faces the same problem: some models use Cifar10 during training (e.g. Stylegan2+ADA+Tune + DI), some don't (e.g. Stylegan2), some may even use different pretraining dataset (e.g. BigGAN, FQ-GAN).
> >
> > I suggest author to provide a clear setting for the methods in Table 3 and Table 4.

---

> > > ### Author Response · Authors · 2023-08-15
> > > **Re-organize Table 3 and Table 4 to make them more clear.**
> > >
> > >
> > > Thank you for your response. Below, we summarize and answer your questions in the following paragraphs.
> > >
> > > **Q1**. More detailed setting of GAN improvement experiment (Table 3 and Table 4).
> > >
> > > **A1**.
> > > First, let us make some clarifications.
> > > By our understanding, we assume that you may have mistaken the "BigGAN+Tune" in Table 3 for our results and thought that we used a BigGAN pre-trained on larger datasets and then finetuned it on CIFAR10.
> > > We want to clarify that we have not experimented with BigGAN and the reported FID for "BigGAN + Tune" is directly taken from Table (b) in Figure 11 of the StyleGAN2-ADA paper [1], where they termed "BigGAN+Tuning".
> > >
> > > StyleGAN-ADA [1] is our main reference paper, and the FID values of ProGAN, AutoGAN, MultiHinge, FQ-GAN, and Stylegan2+ADA+Tune are also taken from the StyleGAN2-Ada paper. **So it is a misunderstanding that we use either different datasets or different methods**.
> > >
> > > Next, for the GAN improvement experiment itself, it is worth emphasizing our goal. We aim to show that by incorporating knowledge of a strong teacher diffusion model, our Diff-Instruct can further improve generators pre-trained with adversarial training (i.e. GAN).
> > > Therefore, taking a pre-trained GAN generator as a baseline, we evaluate whether Diff-Instruct can further improve its generative performance. To make our presentation more clear, we reorganized Table 3 and Table 4 to **Table 3** and **Table 4** to highlight the pre-training and post-training methods.
> > >
> > > **Table3**. Class-conditional sample quality on CIFAR10 with GAN generators. †Models that we implemented. DI represents for Diff-Instruct.
> > > | Method                    | Pre-trained Dataset   | Pre-trained Method    | Post-trained Method   | FID       | IS            |
> > > | :------:                  | :------:              | :------:              | :------:              | :------:  | :------:      |
> > > | BigGAN                    | CIFAR10-Cond          | GAN                   |                       | 14.73     | 9.22          |
> > > | BigGAN+Tune               | CIFAR10-Cond          | GAN                   |                       | 8.47      | 9.07 ± 0.13   |
> > > | MultiHinge                | CIFAR10-Cond          | GAN                   |                       | 6.40      | 9.58 ± 0.09   |
> > > | FQ-GAN                    | CIFAR10-Cond          | GAN                   |                       | 5.59      | 8.48 ± 0.09   |
> > > | Stylegan2                 | CIFAR10-Cond          | GAN                   |                       | 6.96      | 9.53 ± 0.06   |
> > > | Stylegan2†                | CIFAR10-Cond          | GAN                   |                       | 7.01      | 9.23 ± 0.07   |
> > > |**Stylegan2† + DI**        | CIFAR10-Cond          | GAN                   | DI (cond-EDM Teacher) | 6.62      | 9.40 ± 0.06   |
> > > | Stylegan2+ADA             | CIFAR10-Cond          | GAN                   |                       | 3.49      |**10.24 ± 0.07**|
> > > | Stylegan2+ADA+Tune        | CIFAR10-Cond          | GAN                   |                       | 2.42      | 10.14 ± 0.09  |
> > > |**Stylegan2+ADA+Tune + DI**| CIFAR10-Cond          | GAN                   | DI (cond-EDM Teacher) | **2.27**  | 10.11 ± 0.10  |

---

> > > ### Author Response · Authors · 2023-08-15
> > > **Re-organize Table 3 and Table 4 to make them more clear. (2)**
> > >
> > > **Table4**. Unconditional sample quality on CIFAR10 with GAN generators. †Models that we implemented. DI represents for Diff-Instruct.
> > > | Method                    | Pre-trained Dataset   | Pre-trained Method    | Post-trained Method       | FID       | IS            |
> > > | :------:                  | :------:              | :------:              | :------:                  | :------:  | :------:      |
> > > | SNGAN                     | CIFAR10-Uncond        | GAN                   |                           | 21.70     | 8.22          |
> > > | ProGAN                    | CIFAR10-Uncond        | GAN                   |                           | 15.52     | 8.56 ± 0.06   |
> > > | AutoGAN                   | CIFAR10-Uncond        | GAN                   |                           | 12.42     | 8.55 ± 0.10   |
> > > | SNGAN+DGflow              | CIFAR10-Uncond        | GAN                   |                           | 9.35      | 9.62          |
> > > | TransGAN                  | CIFAR10-Uncond        | GAN                   |                           | 9.02      | 9.26          |
> > > | Stylegan2                 | CIFAR10-Uncond        | GAN                   |                           | 8.32      | 9.21 ± 0.09   |
> > > | Stylegan2†                | CIFAR10-Uncond        | GAN                   |                           | 8.21      | 9.09 ± 0.09   |
> > > |**Stylegan2† + DI**        | CIFAR10-Uncond        | GAN                   | DI (Uncond-EDM Teacher)   | 7.56      | 9.16 ± 0.09   |
> > > | Stylegan2+ADA             | CIFAR10-Uncond        | GAN                   |                           | 5.33      |**10.02 ± 0.07**|
> > > | Stylegan2+ADA+Tune        | CIFAR10-Uncond        | GAN                   |                           | 2.92      | 9.83 ± 0.04   |
> > > |**Stylegan2+ADA+Tune + DI**| CIFAR10-Uncond        | GAN                   | DI (Uncond-EDM Teacher)   |**2.71**   | 9.86 ± 0.04   |
> > >
> > > **Analysis**:
> > > As is shown in **Table3** and **Table4**, all generators are pre-trained on the same dataset (CIFAR10 cond/uncond), and **the StyleGAN2 generator performs the best among all generators with adversarial training**. So we decide to further improve the StyleGAN2 generator with Diff-Instruct as post-training to outperform adversarial training by incorporating knowledge of a strong EDM teacher diffusion model. It is worth noting that Diff-Instruct is **data-free**, so the post-training of the pre-trained generator does not need real data (we only need a teacher model). Besides, the Diff-Instruct does not need to sample from the teacher model. This makes Diff-Instruct more efficient than a simple baseline (generates samples from the teacher model and then trains the generator with adversarial training) because generating samples from diffusion models is costly, therefore the simple baseline is costly and can hardly be scaled to large-scale tasks such as text-to-image generation. The results in **Table 3** and **Table 4** show that Diff-Instruct can significantly improve the generators that are trained with adversarial training. For instance, the FID of StyleGAN2+ADA+Tune was improved from 2.42 (which is a previous SOTA for StyleGAN2 generator) to 2.27 for conditional generation, and from 2.92 to 2.71 for unconditional generation.
> > >
> > > In conclusion, in both the diffusion distillation and GAN improvement, the Diff-Instruct has shown its strong empirical performance over baseline methods (models). We really thank you for your valuable suggestions on writing and representation, which help us to improve our work.
> > >
> > > We hope our answers have resolved all your concerns. If you still have any questions, please let us know.
> > >
> > > [1] Training Generative Adversarial Networks with Limited Data

---

> > > > ### Comment · Reviewer_6Ze7 · 2023-08-15
> > > >
> > > > Thanks for you answer. That is clear. I would like to increase my score.

---

> > > > > ### Author Response · Authors · 2023-08-17
> > > > > **Thank you for your constructive suggestions.**
> > > > >
> > > > > We are glad that we have addressed your concerns. Thank you for the advice and we will incorporate them in our revision.

---

### Official Review · Reviewer_62Pb · 2023-07-07

**Soundness:** 4 excellent
**Presentation:** 2 fair
**Contribution:** 3 good
**Rating:** 6
**Confidence:** 4

**Summary:**

The paper introduces a method to transfer the knowledge of a pre-trained diffusion model to an implicit generator model, such that the distribution of the generated samples from the generator matches that of the pre-trained DM. The training is data free, and it can be divided into two alternating phases: training a diffusion model on generated samples from the generator model, and updating the generator model to produce samples that produces similar score with the pre-trained DM.

**Strengths:**

1. The paper tackles a very interesting idea. While there are many papers targeting at improve the sampling speed of diffusion models, and multiple works have designed methods to distill diffusion models, the idea of distilling the knowledge to another generator model is, to the best of my knowledge, novel.

2. The method itself is solid and interesting. With the definition of IKL, the training objective becomes an explicit divergence minimization.

3. Results are comprehensive and strong. It shows results on both distilling diffusion models to a single step and enhancing existing GAN models. It also shows impressive performance on single step generation. The fact that it can further improve SOTA GANs is impressive.



**Weaknesses:**

1. The presentation of the algorithm can be improved. Figure 1 isn't very informative, and a more detailed introduction should be included in the caption. Algorithm 1 isn't very clear, as it only shows the gradient formulation, without introducing how to compute it. For example, I am not very clear about how to estimate the integral in the gradient formulation. Did you just sample one random t as a very rough approximation?

2. As you mentioned, when we assume the generator output has dirac delta distribution, the formulation is essentially the same as SDS in dreamfusion. But we know that GAN is indeed a generator model with dirac delta distribution, so what is the real benefit of assuming non-dirac delta distribution for the generator? Did you actually find any benefit of training with algorithm 1 with an auxiliary DM over just training with 3.3?

3. I do not quite understand the distillation part. When doing the distillation (instead of GAN enhancement), at early stage the generator just output non-sense images, and matching the score on it can be meaningless. Is there any explanation on how it can be applied to distilling DM to a non-pretrained generator?

**Questions:**

See above.

**Limitations:**

Limitations have been discussed.

---

> ### Author Rebuttal · Authors · 2023-08-10
>
> Thank you for your valuable suggestions, we will incorporate them in the revision. In the following paragraphs, we will address your concerns one by one.
>
> **Q1**. The presentation of the algorithm can be improved. I am not very clear about how to estimate the integral in the gradient formulation
>
> **A1**. Thanks. We will revise the representation of Algorithm 1 in the revision. As for the implementation of the gradient estimation, we sample a batch of different times (diffusion sigma levels) and apply forward diffusion on generated images to corresponding times and calculate the gradient with the 2nd step of Algorithm 1. Then we average the gradient over the batch to update the generator. This batch-average manner is the same as training diffusion models.
>
> **Q2**. We know that GAN is indeed a generator model with dirac delta distribution, so what is the real benefit of assuming non-dirac delta distribution for the generator?
>
> **A2**. For a GAN generator, the output distribution is not a single-point distribution, because the input latent vector $z\sim p_Z$ has randomness. In practice, we tried to use equation (3.3) for distillation, but the trained generator shows strong model-collapse behavior, i.e. the generator collapses on certain modes of data, which leads to a distillation failure. The reason behind the failure of using 3.3 other than 3.2 to update the generator lies in that the general generator's output distribution is not single-point, thus using 3.3 is not suitable.
>
> **Q3**. I do not quite understand the distillation part. When doing the distillation (instead of GAN enhancement), at early stage the generator just output non-sense images, and matching the score on it can be meaningless. Is there any explanation on how it can be applied to distilling DM to a non-pretrained generator?
>
> **A3**. We find that by initializing the generator with the teacher diffusion model, the generator's output has a blurred shape of the generated objects and the correct output range (i.e. pixel values in [-1,1]), this makes the learning of score net and updating generator much easier compared to a purely random initialized generator.
>
> Besides, we find that training random-initialized generators also works. But the training takes significantly more time and is relatively unstable when compared with the well-initialized generator. So we choose to initialize the model with pre-trained diffusion. As a simple demonstration, we visualize a batch of class-conditional generated samples from a StyleGAN2 generator that is randomly initialized and trained with Diff-Instruct in **Figure 2** on the additional page in the global author rebuttal cell.
>
> Thank you for your kind suggestions. We hope our answers have resolved your concerns, and if you still have any concerns, please do let us know.

---

### Official Review · Reviewer_Ln4e · 2023-07-07

**Soundness:** 3 good
**Presentation:** 3 good
**Contribution:** 3 good
**Rating:** 4
**Confidence:** 4

**Summary:**

The paper proposes a method for data free knowledge distillation from large scale diffusion models. It tries to distill the knowledge contained into a diffusion model, into another diffusion model or an explicit image generation model like a GAN. They propose a novel divergence measure obtained by integrating the KL divergence along the path of the diffusion model to match distilled model with the original instructor model. They show that the gradient update for the generator can simply be obtained as a function of the score functions as long as the generator is a differentiable function in its output. Using this the paper shows that they are able to distill knowledge from pre-trained diffusion models into generative models like GANs which enables faster inference. The paper provides empirical results to support their claims.

**Strengths:**

1. The paper discusses an important problem of distilling pre-trained diffusion models, especially into single-shot prediction models which enables faster inference.
2. The proposed method is simple to understand and implement in practice.
3. Since the generator is a single-step predictor, it provides extremely fast inference at test time compared to standard diffusion models.

**Weaknesses:**

1. The convergence of the proposed algorithm is not clear in the paper. The authors should elaborate as to why will the generator generate images which resemble the pre-trained diffusion model. In the absence of the generator, one can always sample random images and try to match a new diffusion model to a pre-trained model. Why will this approach not work? (One can use lesser number of reverse diffusion steps for this model)
2. This method requires training two models, which increases the space complexity of training

**Questions:**

1. How does the generator converge to the distribution of the pre-trained diffusion model?
2. Can you reduce the number of reverse diffusion step in the first step of algorithm 1 so that its inference is comparable to the generator? Since the generator indirectly interacts with the pre-trained diffusion model.
3. How will simple score matching work, with lesser number of diffusion steps? What’s the need to introduce a generator model? Is it some form of regularization?

**Limitations:**

1. Authors need to provide an elaborate discussion comparing the proposed method with its baselines for instance consistency models, and justify why the method is better compared to the baseline in terms of the theoretical convergence or inference

---

> ### Author Rebuttal · Authors · 2023-08-10
>
> Thank you for your reviews. We will address your concerns one by one in the following paragraphs. Before that, we first give a summary of the main contributions of our work.
>
> In this work, we introduce the Diff-Instruct, which can utilize a pre-trained teacher diffusion model to train a student one-step generator. To do this, we minimize the Integral KL divergence between the generator's and the teacher diffusion model's distribution. We use an auxiliary score network that is trained to approximate the generator's marginal score functions and used to provide updating gradient for the generator. After training, the auxiliary score net is discarded, and the learned student's one-step generator is used for fast generation.
>
> Next, we address your concerns one by one.
>
> **A1**. (1) We are not sure what you mean by "one can always sample random images and try to match a new diffusion model". We guess you are assuming that the one-step generator is used to help obtain a fewer-step diffusion model, which is similar to what Progressive distillation (PD) does. However, the goal of the Diff-Instruct is to train the one-step generator, instead of a fewer-step diffusion model. So we do not need to sample from diffusion models. To train the one-step generator, we aim to minimize the IKL divergence through Algorithm 1. In algorithm 1, we alternate between training an auxiliary score network $s_\phi$ to learn the generator's marginal score functions and using $s_\phi$ with teacher model $s_{q^{(t)}}$ to provide a gradient for the generator's parameters to minimize the IKL divergence. The algorithm is stable and shows strong empirical performance. If the generator and score network has unlimited expressive ability, then the generator's distribution will converge to the teacher diffusion model's underlying distribution.
>
> (2) Since the one-step generator is what we want and we do not need the diffusion model to sample, we think the concerns in questions, quotes
> <1>"In the absence of the generator",
> <2>"use a lesser number of reverse diffusion steps",
> <3>"What’s the need to introduce a generator model?",
> <4>"Can you reduce the number of reverse diffusion step in the first step of algorithm 1"
> have no relation with Diff-Instruct. I am not sure whether I have addressed your concerns. Please let me know if they are not your concern.
>
> **A2-A5**. We acknowledge that the additional model brings additional memory cost for saving additional models. However, this additional memory cost is limited because the memory bottleneck of training lies in the computational graph of the backpropagation, instead of only saving one more model. In Diff-Instruct, the model $s_\phi$ and the generator $g_\theta$ are updated alternatively, which means that the other model's parameters are fixed and do not participate in back-propagation when one model is being updated. So the memory cost for back-propagating through the computational graph is almost the same as one model. To quantitatively measure how much additional computational costs are brought in, we compare Diff-Instruct and CD's memory and computational costs in **Table 3** in the global author's rebuttal cell.
>
> The result show Diff-Instruct costs almost the same GPU and CPU memory as the baseline CD, but about 1.5 times wall-clock time than the CD for each iteration.
>
> Besides, our empirical evaluation also reveals that the Diff-Instruct converges significantly faster than the baseline method, the consistency distillation (CD). We record the FID curve w.r.t training iterations of Diff-Instruct and CD in the Left part of **Figure 2** on the additional pdf page in the global rebuttal cell. The results show that Diff-Instruct converges significantly faster than CD. For instance, the FID of Diff-Instruct converges to 4.6 with 7k training iterations, while the consistency distilled student model still holds an FID over 30.
>
> **A6** (limitations). Thank you for your reminders. There are mainly four differences (and advantages) between Diff-Instruct and baseline CD.
>
> (1) Diff-Instruct trains the student generator by minimizing the IKL divergence. Instead, the CD trains the student model by minimizing the self-consistency metric. To minimize the self-consistency metric, the CD needs an additional exponential moving average (EMA) model which is slowly updated. Also in practice, this EMA model is hard to tuning to achieve the optimal performance.
>
> (2) The Diff-Instruct allows for a flexible student generator whose input and output dimensions are not necessarily the same. This makes it applicable to transferring knowledge from pre-trained diffusion models to various generators in different applications including diffusion distillation, GAN improvement, and text-to-3D creation. On the contrary, CD requires the student model to have the same input and output dimensions, so CD is not applicable for applications such as GAN improvement and text-to-3D generation.
>
> (3) Diff-Instruct shows better empirical performance (the FID) than CD on ImageNet64 one-step distillation Benchmarks. Besides, its performance does not depend on other learned metrics, such as the LPIPS that is used in CD.
>
> (4) Diff-Instruct is data-free, while CD requires ground truth data. This advantage makes the Diff-Instruct have a wider use than CD, especially for cases when real data is difficult or expensive to obtain.
>
> Despite the advantages, Diff-Instruct has its limitations compared with CD. The CD-distilled model is able to run multiple times to obtain better sample quality, while the Diff-Instruct can only distill to a one-step generator. We think the novel extension of Diff-Instruct to support multiple-step inference is an important future direction.
>
> We hope our answers have resolved your concerns, and if you still have any concerns, please do let us know.

---

> > ### Author Response · Authors · 2023-08-17
> > **Thank your for your reviews, we are happy to provide clarifications.**
> >
> > We sincerely hope that our responses have adequately addressed the concerns you raised in your review. For any unresolved concerns or additional questions, please do not hesitate to let us know. We would be happy to provide further clarification and address any remaining issues.

---

### Official Review · Reviewer_AdFD · 2023-07-13

**Soundness:** 3 good
**Presentation:** 3 good
**Contribution:** 3 good
**Rating:** 8
**Confidence:** 5

**Summary:**

The paper proposes a general framework for distilling a pretrained diffusion model into an arbitrary one-shot latent-variable model. The objective is a reverse KL divergence integrated over time, and an extra score model is learned for the current model distribution. The generator model parameter and the score model parameter are updated alternatively. Empirical results were shown on distilling a diffusion model to a model of same architecture, and finetuning a pretrained GAN model.

**Strengths:**

- This paper proposed a general framework for distilling pretrained diffusion model on a model-architecture agnostic fashion. Meaning that the distilled model doesn't have to have the same model architecture as many other distillation approaches for diffusion model. This can be useful for many potential applications such distilling 2D diffusion models for 3D.
- Introducing a learnable score network for the generator model removes the need of learning an inference network as the usual variational inference requires.
- Overall speaking I think this framework is nice and can potential have impact not only for 2D image distillation / refining, but also cross-modality knowledge transferring.

**Weaknesses:**

- One piece missing and I'm eager to see is, if the implicit model is randomly initialized with an arbitrary architecture (e.g. a styleGAN generator), would this approach work? The experiments show that it works well if the implicit model is either initialized from a pretrained DM or a pretrained GAN model. It makes me doubt if this approach works only when the generator is already well-initialized. Adding this piece will make the paper much more stronger IMO.
-  The approach involves training two models jointly which inevitably introduces the issue of mismatching between two model classes and potentially results in instable training. I'd like to see more analysis towards this issue.

**Questions:**

- Fig.1 is pretty low-res. Consider improving the figure quality.
- There's a concurrent work (not a criticism!): ProlificDreamer: High-Fidelity and Diverse Text-to-3D Generation with Variational Score Distillation, which proposed essentially the same training objective in the context of text-to-3D and justified the objective from W gradient flow. Would be nice to compare with and discuss this work.

**Limitations:**

Yes.

---

> ### Author Rebuttal · Authors · 2023-08-10
>
> Thank you for your valuable suggestions, we will take them in the revision. In the following paragraphs, we will address your concerns one by one.
>
> **Q1**. If the implicit model is randomly initialized with an arbitrary architecture (e.g. a StyleGAN generator), would this approach work?
>
> **A1**. In our exploration, we have tried and found distilling a random initialized generator **also works** for both UNet and StyleGAN generators, but there takes much more time and is relatively unstable than the one that incorporates knowledge of pre-trained diffusion models. Besides, our goal is to transfer as much knowledge as possible from pre-trained diffusion models, so we choose to initialize the generator with the pre-trained model.  As a simple demonstration, we visualize a batch of class-conditional generated samples from a StyleGAN2 generator that is randomly initialized and trained with Diff-Instruct in **Figure 2** on the additional page in the global author rebuttal cell. Thank you for the suggestion, and we plan to add more discussions on training from scratch in our work in the revision.
>
> **Q2**. The approach involves training two models jointly, ... ... and potentially results in unstable training.
>
> **A2**. Thank you for the reminder. First, we want to say that our two models are trained alternatively, instead of jointly. Second, empirically, we find that the training curve of Diff-Instruct in all experiments is quite stable and fast. To demonstrate the comparison, we give **Figure 2** on the additional page in the global author rebuttal cell to show the FID curve with respect to training iterations of Diff-Instruct. The Diff-Instruct's FID on CIFAR10 distillation converges to 4.6 within 7K iterations. We think the reason for such stable training is that the diffusion models have been developed to be able to quickly match the generator's marginal score functions. Besides, we initialize $s_\phi$ with the same weights of teacher model $s_{q^{(t)}}$, which we find helps to stabilize the training than a random initialized $s_\phi$. Since the $s_\phi$ is initialized the same way as $s_{q^{(t)}}$, the gradient for generator $g_\theta$ is small (equation 3.2) at the early stage of the distillation, this makes the $s_\phi$ has sufficient time to match marginal scores of $g_\theta$ distribution.
>
> **Q3**. Fig.1 is pretty low-res. Consider improving the figure quality.
>
> **A3**. Figure 1 is of high resolution, and we guess you are talking about Figure 2. Figure 2 shows generated samples from ImageNet64 and CIFAR10 of 32 resolutions, so their resolutions are low.
>
> **Q4**. There's a concurrent work (not a criticism!): ProlificDreamer, Would be nice to compare with and discuss this work.
>
> **A4**. Thanks for the reminder. It's interesting that ProlificDreamer proposes essentially the same objective as Diff-Instruct in the concurrent work, but we do not know this work before our submission. We plan to add the discussion to this work in our revision. Our paper and ProlificDreamer focus on different aspects. We use Diff-Instruct to obtain state-of-the-art single-step diffusion distillation performance and outperform adversarial training for GAN generators. Instead, the ProlificDreamer specializes in the scenario of text-to-3D creation and shows a really impressive performance. As a brief conclusion, we think both Diff-Instruct and the ProlificDreamer contribute to the research direction that aims to better use the knowledge of pre-trained diffusion models.
>
> Thank you for your kind suggestions. We hope our response has resolved your concerns. If you still have any concerns, please let us know.

---

> > ### Comment · Reviewer_AdFD · 2023-08-14
> > **Thanks for the rebuttal**
> >
> > Thanks for the rebuttal and providing additional results.
> >
> > My questions Q2-Q4 are well addressed by the authors. In terms of Q1, It is a bit surprising to me that this approach can work well with a randomly initialized generator model, as we know the reverse KL in Eq (3.1) is notorious for mode-seeking / mode-collapsing behavior, while seems not the case as shown in new Fig 2 right. Could you comment more on why in your framework, the mode-seeking bad property is avoided? Any special treatment in terms of hyperparamters, training procedure or modeling is required to make it work? Also, could you report numerical evaluations in that case?

---

> > > ### Author Response · Authors · 2023-08-15
> > > **Discussion on mode-seeking issue of random initialized generator: good intuition**
> > >
> > > Thank you for your response. We briefly summarize your question as Q1 and answer it in A1.
> > >
> > > **Q1**. More comment on the mode-seeking issue, training tricks, and numerical evaluation of training from scratch.
> > >
> > > **A1**:
> > > Thank you for your question, we are also interested in the training-from-scratch setting of Diff-Instruct. We agree that reverse KL can potentially lead to a mode-seeking issue, which is a well-known fact in the literature. In our train-from-scratch experiment, we find several "tricks" to help the training.
> > >
> > > (1) **noise level truncation**. We found that when training from scratch, direct use of all noise levels for Diff-Instruct will lead to failure (e.g. the generator fails to generate meaningful contents) because if the generator and teacher model is too dissimilar (as in training from scratch setting), score functions at small noise levels can be uninformative to “pull” the generator to teacher model. So we truncate the noise level by a minimal value (2.0 to 5.0) and find this helps.
> > >
> > > However, we think that truncating noise levels can lead to performance degradation because some noise levels are not used for Diff-Instruct. It is possible that more complex noise annealing tricks can further help but we leave the exploration of them in the revision.
> > >
> > > (2) **larger learning rate for $s_\phi$**. We find it better to use a larger learning rate for the $s_\phi$ than for the generator. Our intuition behind this is to let the generator update a little bit slower so that the diffusion model can be ready to follow the generator.
> > >
> > > **Analysis**:
> > > Overall, training from scratch works but faces a performance drop with an FID of 27.5 and an Inception Score of 9.88, which is significantly worse than the optimal initialized generator (sigma=5.0, FID=4.1).
> > >
> > > We appreciate your keen intuition on mode-seeking behavior for random-initialized generation.
> > > Figure 2 in our added rebuttal page shows some mode-seeking behaviors. For instance, all generated horses seem to turn their head to the right, and the color and shape of generated car images seem not diverse enough. This indicates that randomly initialized generators do demonstrate the **mode-seeking issue**.
> > > Therefore, it highlights the motivation and necessity of our contribution of **applying Tweedie's formula on the teacher diffusion model to obtain a better initialization** of the generator in the main text. We think this initialization technique is novel and easy to use for diffusion distillation, and it can potentially be helpful in other domains of knowledge transfer.
> > >
> > > In conclusion, incorporating knowledge from the teacher model through initialization shows better performance and more stable training than baseline (training from scratch). Developing new techniques, such as introducing a likelihood-ratio estimator, to improve the training from scratch may be an interesting research direction in the future. We thank you for your valuable suggestions which help to improve our work.
> > >
> > > We hope our answers have resolved all your concerns. If you still have any questions, please let us know.

---

> > > > ### Comment · Reviewer_AdFD · 2023-08-15
> > > >
> > > > Thank the authors for clarifying the training details of the training-from-scratch setting, and discussing about the mode-seeking issue. I do agree that this is an interesting and promising research direction to explore, and the results on distilling a well-initialized diffusion model are already pretty convincing. I'd like to increase my score to 8 to support the acceptance of this paper.
> > > >
> > > > Not necessary, but it'd be great if you could expand the discussion on potential new techniques on improving training-from-scratch a bit.

---

> > > > > ### Author Response · Authors · 2023-08-17
> > > > > **Thank you for liking our work. More discussions on training techniques.**
> > > > >
> > > > > We are glad that we have addressed your concerns. Thank you for the advice and we will incorporate them in our revision.
> > > > >
> > > > > We totally agree that training from scratch is an exciting direction. If we can avoid the mode-seeking behavior, training from scratch may potentially lead to stronger performance than the "pretraining + distillation" method.  We think some inspiration can be drawn from two other domains: adversarial training (GAN) and text-to-3D generation.
> > > > >
> > > > > (1) The mode-seeking (or mode-hopping) behavior is a well-known issue for GAN. The community has developed many techniques to address the issue from either theoretical or empirical perspectives. For instance, some divergences, such as the one used in non-saturate GAN [1], have shown promising progress in alleviating mode-seeking behaviors.
> > > > > It would be interesting to explore whether they can help stabilize our Diff-Instruct.
> > > > > Besides, it is well-known that proper regularization of GAN's discriminator (the density ratio estimator) is crucial for the success of GAN training. There is a possibility that some forms of regularization of $s_\phi$ in Diff-Instruct could benefit the training from scratch (currently we do not assign regularization to $s_\phi$).
> > > > >
> > > > > (2) The text-to-3D content creation is another exciting scenario of knowledge transferring from teacher diffusion models to one-step generators. As we have shown in our main text, the Diff-Instruct has laid a solid mathematical foundation for many text-to-3D creation algorithms (such as DreamFusion and its variants) by minimizing IKL divergence. Therefore, techniques from text-to-3D generation algorithms, e.g., annealing the noise schedules of distillation, are potentially useful in other knowledge-transferring applications such as diffusion distillation and GAN improvement.
> > > > >
> > > > > [1] Generative Adversarial Networks
> > > > >
> > > > > [2] Towards Principled Methods for Training Generative Adversarial Networks

---

### Author Rebuttal · Authors · 2023-08-10

Thank all reviewers for your valuable feedback. In the rebuttal period, we run an additional experiment to compare the computational and memory costs of Diff-Instruct with our baseline method Consistency Distillation in Table 3. The test was run on 2 Nvidia V100 GPUs with 128 batch size and PyTorch distributed data-parallel mechanism.

**Table3**. Comparison of Memory Costs and Wall-Clock Timing of CD and Diff-Instruct. The Peak GPU-Memo shows the Maximum observed GPU memory caused by the Tensor and Computational Graph of the host GPU. Sec-per-K Iterations report the wall-clock time for each training iteration. For Diff-Instruct, each iteration consists of two stages as in Algorithm 1 in the main text. Test environment: PyTorch 1.12.1 and Torchvision 0.13.1, and Torch.distributed.parallel on 2 V100 GPUs.
| Method        | Peak GPU-Memo(GB) | Peak CPU-Memo(GB) | Sec-per-K Iterations  |
| :------:      | :------:          | :------:          | :------:              |
| CD            | 9.55              | 2.75              | 0.0489                |
| Diff-Instruct | 10.40             | 2.78              | 0.0728                |

The result shows that Diff-Instruct brings in minor additional memory costs than CD (10.40 over 9.55). This is because the Diff-Instruct only needs additional GPU memory to save the auxiliary model $s_\phi$. But the $s_\phi$ and generator $g_\theta$ are updated alternatively, so their computational graph does not interacts. As a result, the memory bottleneck caused by computational graph and back-propagation does not bring more costs to Diff-Instruct.

As for the wall-clock time for 1K iterations, we see Diff-Instruct costs 0.0728 seconds, while the CD costs 0.0489 seconds. This is because each iteration of Diff-Instruct consists of two alternate steps as we show in Algorithm 1. Overall, the Diff-Instruct costs almost the same GPU and CPU memory as the baseline CD, but about 1.5 times wall-clock time than the CD for each iteration.

Besides the computational costs, we also show the convergence of FID of Diff-Instruct and CD in the left part of **Figure 2** on our additional page of figures. In that Figure, the Diff-Instruct's FID on CIFAR10 distillation converges to 4.6 within 7K iterations, while the CD distilled model's FID still remains larger than 30. This shows the Diff-Instruct converges quickly with iterations.

---

### Decision · Program_Chairs · 2023-09-21

**Decision:**

Accept (poster)

**Comment:**

This paper has garnered mostly positive reviews, while the main concerns raised in the reviews (having mostly to do with the convergence of the algorithm and the relation to few-step diffusion models) seem to have been properly addressed in the rebuttals.
Reviewers appreciated the clear motivation, the formal foundation of the proposed method and the clarity of its implementation.
Reviewers were also positively impressed expressed by the comprehensive and convincing experimental results validating the method.
Finally, reviewers concluded that the impact of this work might even reach beyond distilling models for image generation, and extend to cross-modality applications.